# What does the impurity variability at the microscale represent in ice cores? Insights from a conceptual approach

Piers Larkman[1], Rachael H. Rhodes[2], Nicolas Stoll[1], Carlo Barbante[1,3], and Pascal Bohleber[1,4]

[1]Department of Environmental Sciences, Informatics and Statistics, Ca'Foscari University of Venice, Venice, Italy
[2]Department of Earth Sciences, University of Cambridge, Cambridge, United Kingdom
[3]Institute of Polar Sciences, National Research Council (CNR-ISP), Venice Mestre, Italy
[4]Department of Geosciences, Alfred Wegener Institute Helmholtz Centre for Polar and Marine Research, Bremerhaven, Germany

**Correspondence:** Piers Larkman (piersmichael.larkman@unive.it)

**Abstract.** Measuring aerosol-related impurities in ice cores gives insight into Earth's past climate conditions. In order to resolve highly thinned layers and to investigate post-depositional processes, such measurements require high-resolution analysis, especially in deep ice. Micron-resolution impurity data can be collected using laser ablation inductively coupled plasma mass spectrometry (LA-ICP-MS) but this requires careful assessment to avoid misinterpretation. 2D imaging with LA-ICP-MS has provided significant new insight, often showing an association between soluble impurities and the ice crystal matrix, but interpreting 1D signals collected with LA-ICP-MS remains challenging partially due to this impurity-boundary association manifesting strongly in measured signals. In this work, a computational framework has been developed integrating insights from 2D imaging to aid the interpretation of 1D signals. The framework utilises a simulated model of a macroscopic ice volume with a representative microstructure and soluble impurity localisation that statistically represents distributions seen in 2D maps, allowing quantitative assessment of the imprint of the ice matrix on 1D signals collected from the volume. Input data were collected from four ice core samples from Greenland and Antarctica. For the samples measured, quantifying the variability of 1D signals due to the impurity-matrix imprint shows that modelled continuous bulk signal intensity at the centimetre scale varies below 2 % away from an idealised measurement that captures all variability. In contrast, modelled single-profile micron-resolution LA-ICP-MS signals can vary by more than an average of 100 %. Combining individual LA-ICP-MS signals into smoothed and spatially averaged signals can reduce this variation to between 1.5 and 5.9 %. This approach guides collecting layer-representative signals from LA-ICP-MS line profiles and may help to bridge the scale gap between LA-ICP-MS data and data collected from meltwater analysis.

# 1 Introduction

Ice cores collected from Earth's polar regions contain invaluable information relating to its climate system, with continuous records reaching back as far as 800,000 years (Loulergue et al., 2008; Brook and Buizert, 2018). Analysis of well-preserved old ice, such as that targeted in the Beyond EPICA drilling on the Antarctic Plateau, aims to extend this record back to approximately 1.5 million years (Chung et al., 2023). Ice sections originating from near the bottom of ice sheets, including that targeted for the Beyond EPICA core, contain very thinned layers, with many thousands of years of climate information compressed into small vertical sections. Such ice will have undergone significant post-depositional changes.

A subject of interest within these cores are the aerosol-related impurities in the ice (e.g. Legrand and Mayewski, 1997), which can be used as a proxy to reconstruct past climate conditions over timescales ranging from seasonal to millennial. A widely employed technique for collecting such signals is continuous flow analysis (CFA), which outputs a one-dimensional (1D) impurity signal along the down-core axis at centimetre depth resolution (Kaufmann et al., 2008). An example target impurity is sodium, for which potential links to sea ice extent are discussed (Abram et al., 2013). As there is likely more than 14 ka of ice per metre in the deep ice of the Beyond EPICA core, high-resolution analysis is key to deciphering climate signals in these highly thinned sections. Such analysis will require resolutions beyond that delivered by CFA and also careful assessment of the impact of post-depositional changes of impurity localisation.

To measure impurity signals at micron-resolution, laser ablation inductively coupled plasma mass spectrometry (LA-ICP-MS) has been applied to ice core analysis (Reinhardt et al., 2001). Ablating ice in its solid form, LA-ICP-MS preserves information on impurity location in the ice matrix while analysing the surface of the sample (Müller et al., 2011). Two-dimensional (2D) state-of-the-art imaging of impurities using LA-ICP-MS has shown that the location of (mostly soluble) impurities, such as sodium and magnesium, can significantly correlate with the location of boundaries between crystals in the ice matrix (Stoll et al., 2023; Bohleber et al., 2020). This impurity-boundary association imprints onto 1D line profile signals collected along the down-core axis of samples, changing the resultant signal depending on the lateral position on the ice the signal is collected from (Bohleber et al., 2021). It is now clear that this imprint obscures the interpretation of such profiles in the context of extracting a climate signal, but the extent to which this occurs will depend on factors such as the degree of impurity localisation, which can vary between elemental species, and grain size.

The micron-resolution 2D data sampled from the surface with LA-ICP-MS greatly differs in nature from the centimetre-resolution 1D bulk impurity data obtained with CFA, producing a scale and dimensional gap between their outputs. It remains unclear how LA-ICP-MS signals collected from ice core samples containing a stratigraphy that encodes climate variability should be interpreted, partially due to the impurity-boundary association showing up in these micron-resolution measurements. Despite methodological differences in LA-ICP-MS and CFA, a phenomenological link has been made between 1D down-core signals collected using LA-ICP-MS and CFA (Della Lunga et al., 2017; Spaulding et al., 2017), after applying heavy smoothing to LA-ICP-MS signals. A deeper explanation of this link between the two techniques must come from an improved understanding of the chemical signals in ice across different length scales.

To allow exploration of how impurity localisation, and therefore factors such as climate period and grain size, impacts measured signals, a computational framework that allows extensive analysis of LA-ICP-MS and CFA data has been developed. This open-source framework[1] developed in Python is designed to guide experimental data collection, especially when attempting to capture layer signals with 1D LA-ICP-MS profiles. Generating a computational model of a macroscopic ice volume, comparable to the dimensions of a sample melted during CFA, that is statistically representative of grain and impurity properties revealed by 2D LA-ICP-MS imaging allows us to contrast modelled and empirical LA-ICP-MS data. This delivers insight into how the spatial distribution of soluble impurities impacts signal collection, assists in bridging the scale and dimension gap between LA-ICP-MS and CFA measurements, and allows studies that are not easily possible with empirical measurements. Presenting this new conceptual approach, this paper aims to:

– Present 1D profiles and 2D intensity maps collected using LA-ICP-MS from sections of Antarctic and Greenland ice cores. Focusing on the mostly soluble impurities we take sodium as an archetypal species.

– Outline the theoretical foundation, computational implementation, and validation of a framework based on a three-dimensional (3D) model that captures the localisation of soluble impurities in ice at the microscale while being statistically representative at the macroscale.

– Establish an initial application of this framework, analysing sodium as an archetypal soluble impurity mainly distributed at grain boundaries, to investigate how the spatial distribution of soluble impurities impacts the representativeness of high-resolution centimetre length 1D signals taken along the down-core axis.

Data are measured and analysed from Holocene and Last Glacial Period (LGP) sections of the Antarctic EPICA dome C (EDC) (Stauffer et al., 2004) and Greenland Renland Ice Cap Project (RECAP) (Simonsen et al., 2019) ice cores. The discussion of these data demonstrates the framework in relatively shallow ice sections, targetting soluble impurities, from which developments can be made to investigate deep ice sections and insoluble impurities.

## 2 Methods

### 2.1 Overview

The developed framework's inputs, operation, and outputs are visualised in Fig. 1. Optical and chemical data are collected experimentally from ice samples using an LA-ICP-MS system to form an empirical input for the framework. These data reveal the spatial distribution of soluble impurities, which are referred to interchangeably with 'impurities' throughout this study, i.e., their localisation at the grain boundaries. The impurity distribution is combined with mean grain size measurements to parameterise the generation of a 3D model representing a macroscopic volume of an ice sample.

This 3D model captures the structure and impurity distribution of measured ice samples. The imprinted impurity distribution is unchanging with depth, which, if a climate signal is considered to be present in an ice sample as a sequence of discrete

---

[1]Code available at https://github.com/Piers-Larkman/Ice_Impurities

constant values, represents a simple manifestation of a climate signal. The conditions under which this signal can be reliably extracted are investigated, and the conclusions are extended to guide and interpret empirical analysis. More complex climate signals can be constructed within the modelled ice, although the mode of this climate signal should not alter the interpretation of the present discussion.

The framework utilises a 3D model to allow both 1D signals representing LA-ICP-MS and bulk CFA measurements to be simulated along the down-core axis of the modelled volume by recording, combining, and processing the intensity at each point in a vertical profile. Utilising the fact that the climate signal present in the modelled volume is an un-changing mean intensity, these signals are then analysed to understand how well they capture this underlying signal.

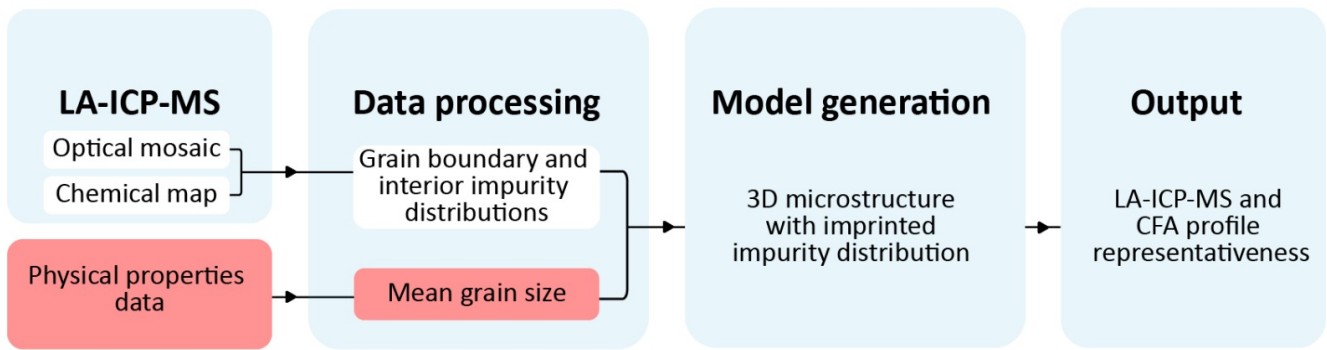

**Figure 1.** Flow chart detailing the framework operation. The mean grain size data for EDC and RECAP are from EPICA community members (2004) and Weikusat et al. (2024), respectively.

## 2.2   Sample selection

Samples were selected to cover a broad range of conditions, including both Greenland and Antarctica as well as glacial and interglacial periods. Four ice samples were analysed and modelled, two from the EDC ice core (Stauffer et al., 2004) and two from the RECAP ice core (Simonsen et al., 2019) (Table 1). Ages for EDC samples are from the AICC2023 timescale for the EDC ice core (Bouchet et al., 2023) and from the RECAP time scale for RECAP samples (Simonsen et al., 2019) and show samples originate from either the Holocene or LGP. Grain radius data are taken from published values (EPICA community

members, 2004; Weikusat et al., 2024).

## 2.3   Experimental

### 2.3.1   Data collection

The LA-ICP-MS setup at the University of Venice was used adhering to current best practice for analysis on ice (Bohleber et al., 2024). The setup utilises an Analyte Excite ArF excimer 193 nm laser with a HelEx II two-volume ablation chamber

(Teledyne CETAC Photon Machines) connected to an iCAP-RQ quadrupole ICP-MS (Thermo Scientific) using a rapid aerosol

**Table 1.** Information on all analysed samples. Sample depth is the top of the sample, all relative depths discussed in the paper are reported with reference to this top depth. Grain radius is the mean effective spherical grain radius at the reported depth. The lateral separation of profiles is their separation measured perpendicular to the down-core axis and is illustrated in Fig. 6 (c).

| Ice core | EDC | EDC | RECAP | RECAP |
|---|---|---|---|---|
| Climate period | Holocene | LGP | Holocene | LGP |
| Top depth (m) | 282.23 | 1096.45 | 495.18 | 536.70 |
| Age (yr b1950) | 9000 | 75000 | 5800 | 35000 |
| Sample length (mm) | 80 | 79 | 80 | 59 |
| Mean grain radius (mm) | 1.3 | 2.3 | 4.2 | 1.7 |
| Number of LA-ICP-MS profiles measured | 10 | 10 | 4 | 6 |
| Profile lateral separation ($\mu$m) | 80 to 6000 | 80 to 12000 | 1000 to 5000 | 1000 to 5000 |

yr b1950: years before 1950 CE

transfer line. Samples were prepared with a thickness of approximately $1\,\mathrm{cm}$, a width of $2\,\mathrm{cm}$ and lengths reported in Table 1. During analysis, samples were held at a stable temperature of approximately $-23\,°\mathrm{C}$. An optical mosaic of the surface of each sample was taken using an integrated optical camera. Impurity data, including sodium, were recorded as 1D lines and 2D maps using the laser with spot size $40\,\mu\mathrm{m}$, firing rate $300\,Hz$, and a fluence of $3.5\,J/cm^2$. After collection, uncalibrated intensity data from the ICP-MS were corrected for background effects and drift using the software HDIP (Teledyne CETAC Photon Machines), which was also used to create impurity maps.

## 2.4 Computational framework

The computational framework does not aim to replicate the physical processes involved in grain growth and impurity localisation but to create a statistically representative microstructure and associated soluble impurity distribution. Its construction breaks down into the following steps.

### 2.4.1 LA-ICP-MS data processing

The grain boundary network was identified manually in the impurity maps through comparison with optical images. Pixels of high intensity in the chemical map which were located at visible grain boundaries in the optical image were considered as grain boundary pixels. High-intensity pixels away from boundaries were considered to be impurities localised at dust particles and are treated as grain-interior pixels. This approach results in a binary mask which was applied to chemical maps to separate grain boundary and grain interior pixels. The intensities of these pixel classes was then recorded and turned into a probability distribution capturing the probability that a given pixel has a certain intensity.

### 2.4.2 Ice structure generation

A 3D Poisson Voronoi tessellation (Zheng et al., 1996) is used to create the structure of modelled ice volumes. Voronoi tessellations are produced by seeding region centres in a space, at random locations in the case of Poisson Voronoi tessellations, and allowing the regions to grow until they intersect with a neighbouring region. At this intersection, a boundary between the regions is formed. The region shapes are governed by how distance is measured in the space. The generalised distance formula in three dimensions allows calculation of the distance, D, between two points, $\mathbf{x} = (x_1, x_2, x_3)$ and $\mathbf{y} = (y_1, y_2, y_3)$

$$D(x,y) = \left( \sum_{i=1}^{3} |x_i - y_i|^p \right)^{\frac{1}{p}}. \tag{1}$$

Where $p = 2$ the resulting distance is the Euclidean distance. Changing $p$ produces different shaped grains. This process produces notional spaces with regions classified either as region (grain) interiors or region (grain) boundaries.

To match the average grain radii of a target ice sample, a suitable number of grains are seeded to create regions with the same 3D grain-number density as the physical sample, that is the same number of grains per unit volume. This process results in a space containing grains with a grain volume distribution that conforms to a gamma distribution (Ferenc and Néda, 2007), which is parameterised in the supplementary material to this paper, with a mean grain radius the same as the target ice sample. To create a spatial link to the pixels of the impurity maps, the Poisson Voronoi tessellation is built in a volume comprised of voxels, the extension of pixels to three-dimensions. The modelled volume is completely populated by voxels assigned to grain interior or boundary regions as illustrated in Fig. 2. The model treats voxels as having an edge length corresponding to the pixel edge length and, therefore, the laser spot size of the LA-ICP-MS map which it represents, $40\,\mu m$ in the case of the maps collected as described in Sect. 2.3. This allows the dimensions of the notional volume to be tied to the dimensions of physical ice samples. Each voxel has a coordinate (x,y,z), with the X, Y, Z coordinate system illustrated in Fig. 2. The modelled space is taken to have the Z axis aligned with the down-core axis of modelled samples.

### 2.4.3 Impurity distribution imprint

Each voxel in the generated space is assigned a numerical value representing its impurity intensity. This value is assigned by taking the two probability distributions from empirical LA-ICP-MS mapping described in Sect. 2.4.1, one for grain boundaries and another for grain interiors, and drawing a random value from these distributions for each voxel, depending on its classification as grain or boundary. The resulting intensity distribution resembles the intensity representation shown in Fig. 2.

### 2.4.4 Simulating and combining signals

Simulated LA-ICP-MS signals are obtained by recording the voxel intensity at each Z position of a profile of voxels, resulting in a 1D signal at $40\,\mu m$ resolution which runs the entire Z-axis of the modelled volume. Fig. 2 illustrates the paths of two such

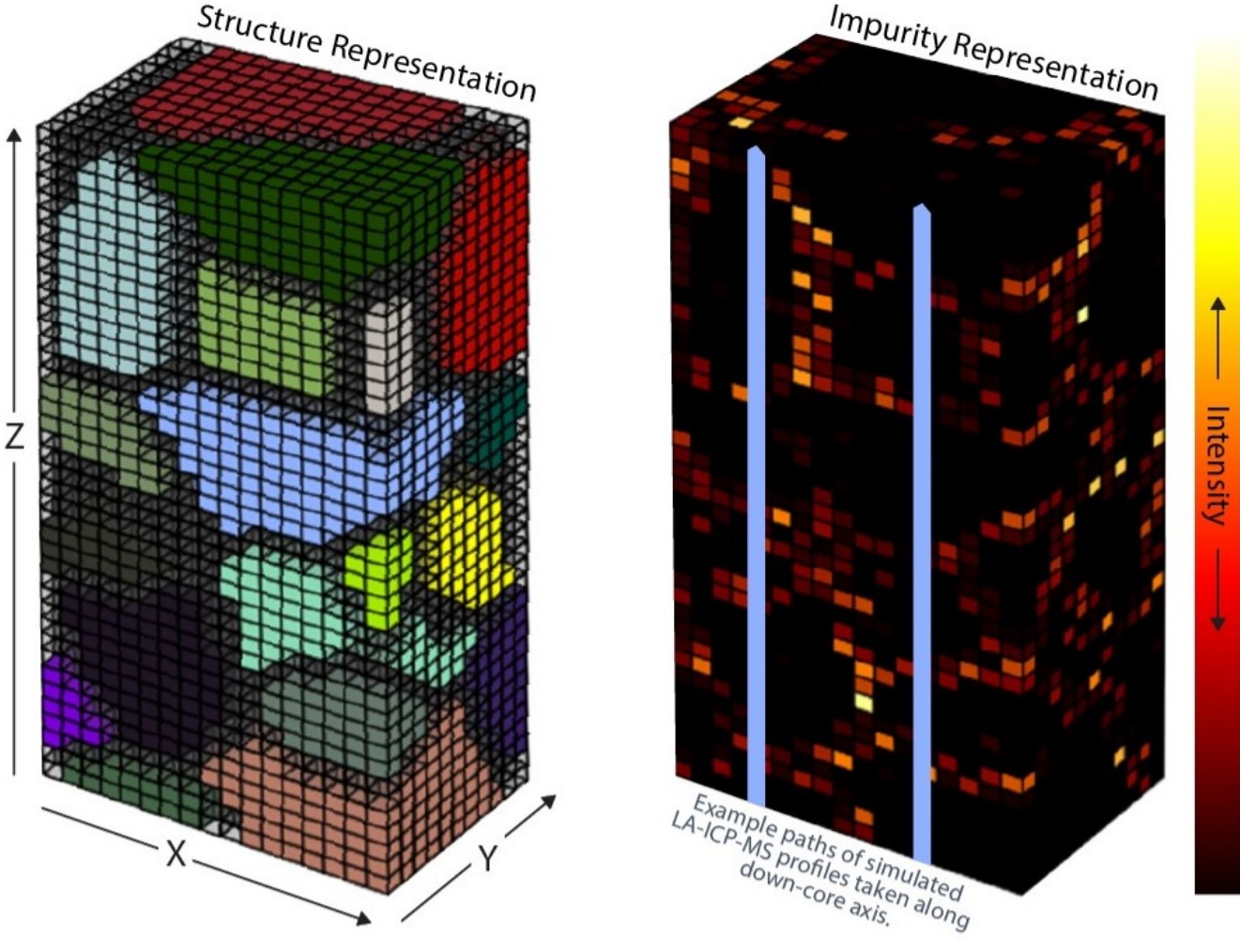

**Figure 2.** Visual representation of a 3D Poisson Voronoi tessellation used to represent a polycrystalline material. Each voxel, represented by a coloured cube, in the space is considered to have an edge length corresponding to the laser spot size used to measure modelled samples. The representation on the left shows the grain boundary voxels in transparent grey and each grain as a different colour. The representation on the right shows the intensity distribution imprinted on the same volume. Volumes generated for analysis are much larger than this small illustrative example, which, given a 40 μm voxel size, represents a 0.60 by 0.44 by 1.8 $mm$ volume.

parallel profiles as blue arrows. Intensity signals from directly adjacent profiles can be summed to create a signal simulating LA-ICP-MS data collection carried out with a larger spot size. Spatially averaged signals can be produced by taking the average of two or more adjacent or non-adjacent single-profiles signals. For spot sizes larger than $40\,\mu m$, all simulated LA-ICP-MS signals are smoothed using a 1D Gaussian kernel with a standard deviation, $\sigma$, set to the laser spot size, following the procedure described in Bohleber et al. (2021). Additional subsequent Gaussian smoothing can then be applied as a post-processing step. Simulation of a CFA-like signal is only possible in a 3D model and is calculated by summing the impurity values of all the voxels in each Z plane and applying Gaussian smoothing with a one-centimetre wide kernel. This approximates the collection of a smoothed bulk signal resulting from experimental CFA (Erhardt et al., 2023), without considering effects such as dispersion (Breton et al., 2012).

## 2.5    Modelled data analysis

These modelled signals were then analysed to give insight into how the underlying impurity distribution creates variability in measurement. It is assumed that centimetre-scale bulk volumes of ice have an invariant intensity distribution in the X and Y directions. This bulk-invariance also holds in the Z direction of modelled ice. This Z-invariance can be interpreted as an ice sample with an unchanging climate signal despite micro-scale variability in the spatial distribution of sodium arising from the impurity-boundary association. The bulk-invariant impurity distribution in all directions means that the mean average intensity in the space, $\bar{I}$, serves as a reference value: the intensity value that would be recorded if the entire volume were melted and measured. If some sub-volume of the modelled space is representative of the volume as a whole, it will have a mean intensity of $\bar{I}$. Therefore, if a single-profile laser signal, a spatially averaged signal resulting from the combination of several profiles, or the simulated CFA signal has an average intensity approaching $\bar{I}$ at each Z value, the signal can be considered representative at every depth. A metric to measure how much the spatial distribution of impurities affects some signal, $I(z_i)$, or the signal representativeness, is its mean absolute deviation (MAD) from $\bar{I}$. For a signal of length $l$, the MAD measured in intensity units, $MAD_I$, is calculated as

$$MAD_I = \frac{1}{l}\sum_{i=1}^{l}|I(z_i) - \bar{I}|. \tag{2}$$

To allow easy comparison of variation between ice samples, MAD values calculated using equation 2 are reported as percentages normalised to $\bar{I}$

$$MAD = \frac{MAD_I}{\bar{I}} \times 100. \tag{3}$$

A MAD of 0 % as calculated using equation 3 represents a signal that fully captures the underlying intensity distribution at every depth interval.

## 3 Results

 ### 3.1 Experimental LA-ICP-MS

All measured samples' optical and intensity maps are shown in Fig. 3, alongside the grain boundary segmentation used to isolate the grain interior and boundary intensities. A comparison of the optical images and intensity maps in Fig. 3 shows sodium is concentrated preferentially at the grain boundaries compared with grain interiors for all measured samples. This bimodal distribution is evident in Fig. 4, which shows the resulting frequency-normalised impurity distributions acquired from overlaying the boundary segmentation mask onto the intensity maps for each measured sample. All samples have higher average intensities at grain boundaries than in grain interiors. Pixels below the detection limit of the ICP-MS have their intensities recorded as zero after drift and background correction. Note that different intensity plots are not directly comparable as no calibration was performed.

Individual, spatially averaged, and smoothed signals collected from the EDC Holocene sample are shown in Fig. 5. Equivalent figures for all other measured ice samples are shown in the supplementary material. Similar to what was noted in previous studies (Bohleber et al., 2020; Della Lunga et al., 2017), experimentally collected signals vary significantly when collected at different lateral positions on the ice surface due in part to the association of impurities with grain boundaries. The two signals plotted in Fig. 5 (a) for the EDC Holocene sample are laterally separated by $160\,\mu m$. Even at this short distance, the signals have different numbers of peaks at varying positions and intensities. The spatially averaged signal in Fig. 5 (b) averages these differences somewhat, lowering overall intensity variations (note the different y-axis scale). This averaging and smoothing is further evident in Fig. 5 (c), a smoothed version of the data in (b).

### 3.2 Computational

Parameterised by the grain radii reported in Table 1 and impurity distributions in Fig. 4, modelled representations of ice microstructure and impurity distribution were produced for all the analysed ice sections. All samples are modelled to have a cross-sectional area of 1 by $2 \pm 0.2$ cm with lengths corresponding to the sample lengths from Table 1. This was computationally more efficient than generating a full 3.5 by 3.5 $cm$ cross-section typically used in CFA, although no principal limitation prevents simulation of a volume with this cross-section. Both EDC and the RECAP LGP samples were generated in approximately half a day using a laptop computer. The RECAP Holocene sample required more RAM to manage larger grain sizes and took 4 days to generate on a high-performance computing system. To illustrate the extremes in grain sizes, one face from each of the 3D-modelled EDC Holocene and RECAP Holocene samples are shown in Fig. 6. The simulated LA-ICP-MS signals plotted for EDC Holocene in Fig. 7 originate from profiles taken along the face shown in Fig. 6 (a). Equivalent figures for all other modelled ice volumes are contained in the supplementary material.

Figure 7 shows signals collected under different simulated conditions, with intensity values normalised such that $\bar{I}$ has an intensity of 1. Figure 7 (a) shows two modelled signals, collected using a $40\,\mu m$ laser spot, that are separated by $160\,\mu m$, representing the modelled equivalent of the two profiles in Fig. 5 (a). The modelled LA-ICP-MS signals show the same

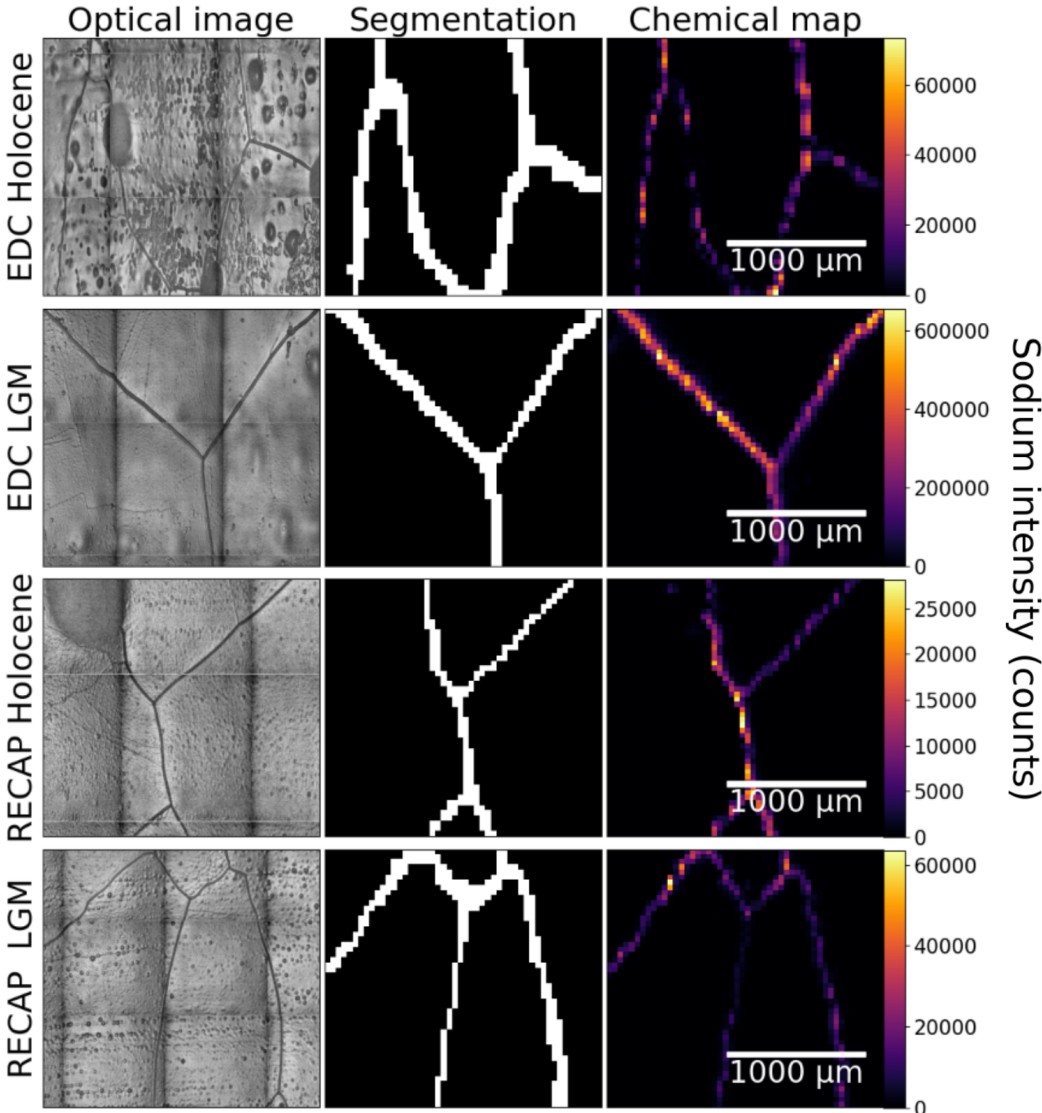

**Figure 3.** Columns show optical image (left), grain boundary segmentation (middle), and sodium intensity map (right) for all of the analysed samples, one sample in each row. The dark grid visible in the optical mosaics is an imaging artifact. Grain boundaries are visible as dark lines in the optical image, and bubbles as dark rounded regions. Each intensity map has its own intensity scale. The spatial scale bar relates to all images for each sample. The areas shown in this image are small snapshots, with the full data shared in the repository associated with this work.

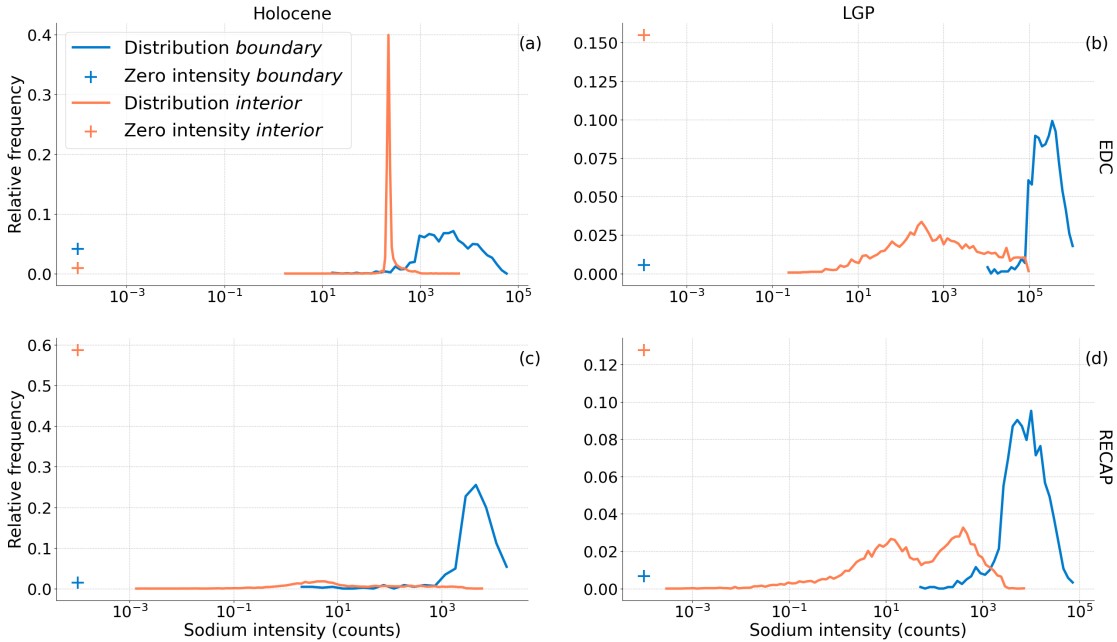

**Figure 4.** Sodium intensity probability distributions for all measured samples. These distributions result from the normalisation of distributions acquired from the application of binary masks which separate grain boundaries from grain interiors to the intensity maps, both shown in Fig. 3. The zero intensity cross represents pixels with intensities below the detection limit of the ICP-MS. The legend applies to all plots.

general features as experimentally measured LA-ICP-MS signals, with large spikes in intensity where profiles intersect grain boundaries. Figure 7 (b) shows the result of simulating two profiles, centred at the same point as those in (a), with a spot size of $120\,\mu m$, comparable to the $100\,\mu m$ spot size used in previous studies (Spaulding et al., 2017; Sneed et al., 2015), which show less variation around $\bar{I}$. The spatially averaged signal resulting from combining all $40\,\mu m$ profiles along the illustrated modelled face, shown in Fig. 6 (a), is plotted in Fig. 7 (c) and its CFA-resolution smoothed equivalent in (d). These signals represent the largest amount of data which can be collected if limited to measuring the surface of only one face of ice samples during LA-ICP-MS, which is a common restriction for such analysis. The simulated CFA signal is plotted in (e) and shows the least variation around the mean of all simulated signals. The large smoothing applied to the signals plotted in (d) and (e) reduces the variation around $\bar{I}$ to the order of 2 % or less. In the illustrated case, both signals are similar in the sense that they show very little variation around $\bar{I}$, although at the narrow range of intensity values in this plotted data, the signals show roughly opposite trends. Data for the other modelled samples contained in the supplementary material show both similar and dissimilar trends, highlighting this as a product of the narrow y-scale used for these plots.

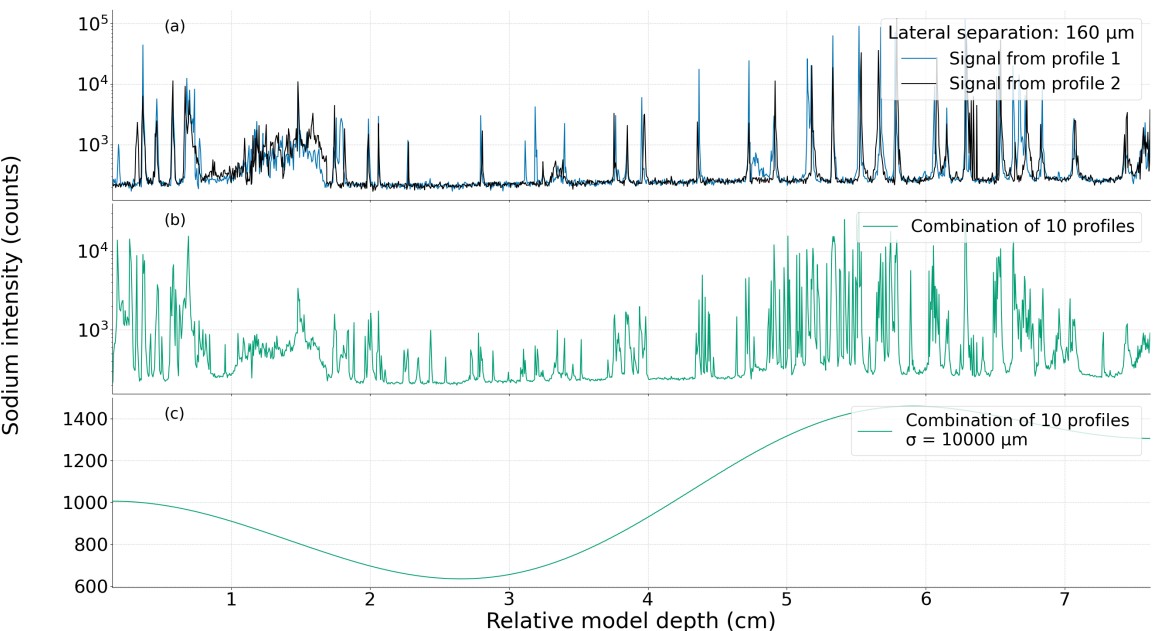

**Figure 5.** Measured LA-ICP-MS signals resulting from line profiles taken across the surface of the EDC Holocene sample. All profiles run down the central core axis. Panel (a) shows two signals resulting from two parallel laser tracks. Panel (b) shows the spatially averaged signal resulting from combining all measured parallel profiles, including the two signals in (a), with a range of separations between adjacent profiles. Panel (c) shows this spatially averaged signal after smoothing to a 1 cm resolution and has a linear y-axis, as the variations at this level of smoothing are relatively small.

MAD values are visualised for data collected from modelled EDC Holocene and RECAP Holocene ice in Fig. 8 with key values for all ice samples reported in Table 2. Figure 8 shows how the MAD for different signals changes based on how many profiles are combined to construct a spatially averaged signal and how much smoothing is applied. Data for the EDC Holocene sample is shown in (a) and (b) and the RECAP Holocene sample in (c) and (d). Panels (a) and (c) show data for $40\,\mu m$ spot size signals and (c) and (d) for $280\,\mu m$ spot size. The general trends are that (1) MAD decreases asymptotically as more profiles are averaged, (2) smoothing reduces signal MAD by some constant, regardless of the number of profiles a spatially averaged signal comprises, and (3) larger spot sizes produce signals with smaller MADs. These general trends hold for all measured ice intervals. For the EDC Holocene ice, single profile signals taken at $40\,\mu m$ have MADs of over 100 %, meaning that signal intensities can vary by over 100 % of the mean intensity in the space. By comparison, the simulated CFA signal, which gives the most representative signals of those simulated, has a deviation of 0.7 %. Spatially averaged signals constructed from 10 profiles, such as the signal experimentally measured and plotted in 5 (b), show variations on average of 62 % for EDC

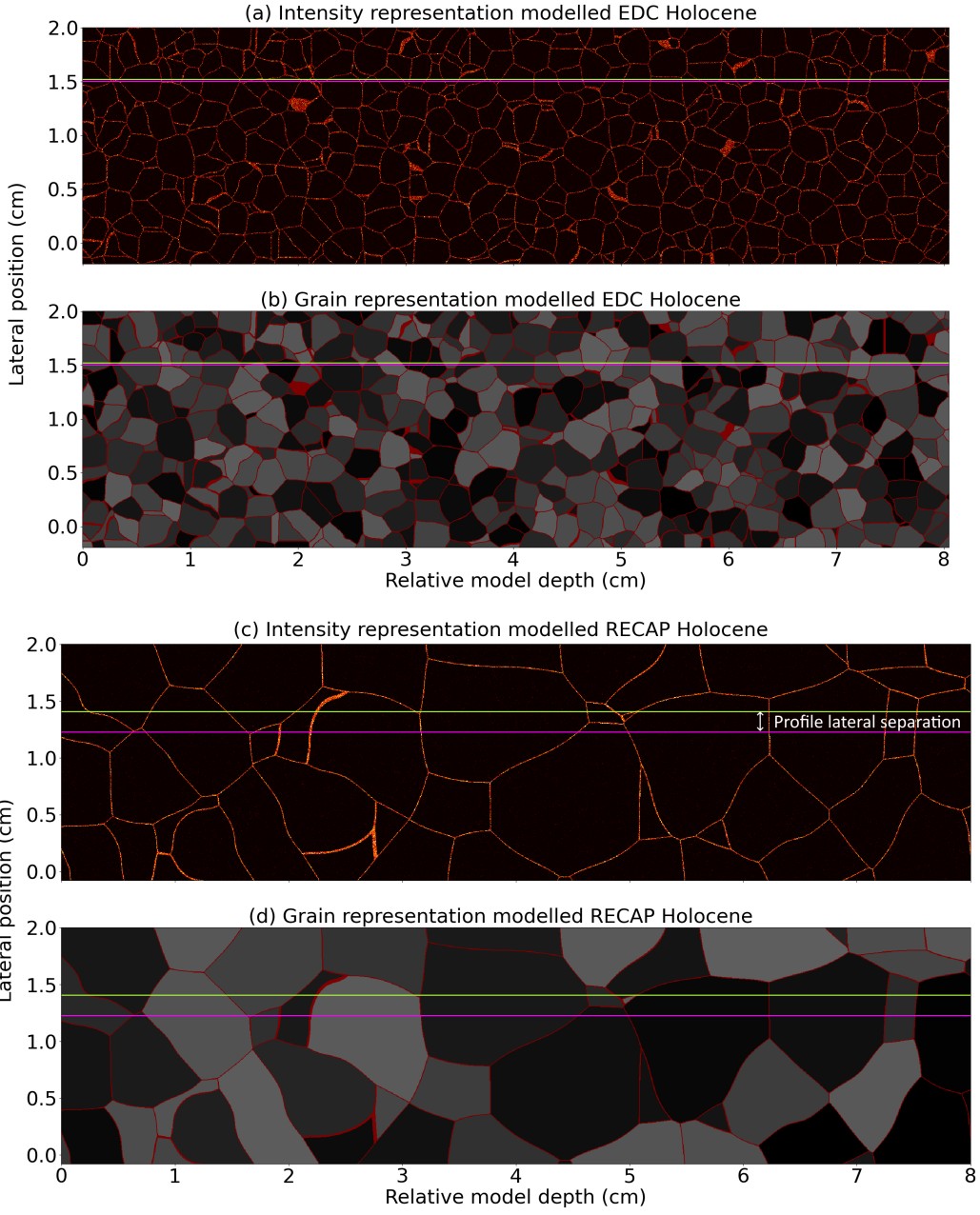

**Figure 6.** The intensity, (a) and (c), and structural, (b) and (d), representations of one modelled face of the EDC Holocene and RECAP Holocene samples. The structural representation shows grains as different shades (which do not hold any special significance), separated by grain boundaries represented in red. The colour scale for the intensity representations has been adjusted for readability and holds the general trend of brighter colours showing greater intensities, as used in figure 2. Each of the rows in the intensity representation can be taken as a separate laser profile. The green and magenta lines in panels (a) and (b) show the track of the profiles plotted in Fig. 7 for the EDC Holocene sample, separated by the lateral separation reported in Table 1. This separation is illustrated in (c).

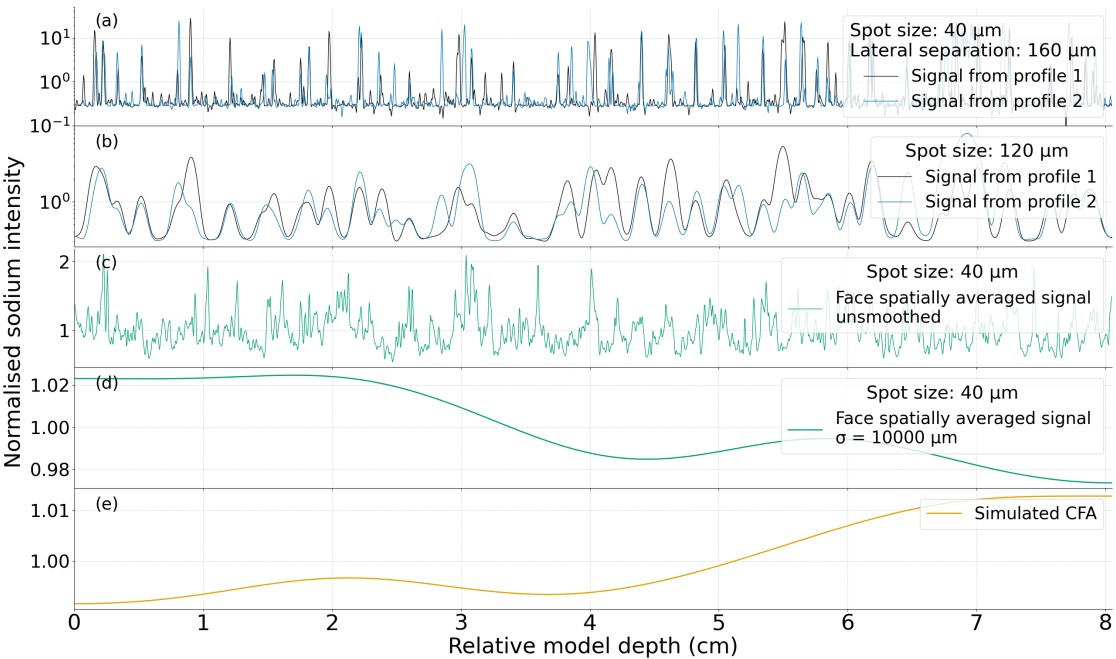

**Figure 7.** Line profile signals for the modelled EDC Holocene ice normalised by dividing by the volume average intensity, $\bar{I}$. Panel (a) shows signals acquired from $40\,\mu m$ spot size profiles taken from the tracks indicated in Fig. 6 and is the modelled equivalent of the plot in Fig. 5 (a). Signals resulting from simulating a $120\,\mu m$ spot size along these profiles are shown in (b). The resulting signal from combining all possible profiles from the face in Fig. 6 is shown unsmoothed in (c) and smoothed to CFA resolution in (d). The simulated CFA signal is plotted in (e). Note the different y-axis scales for each panel, particularly the small range used in (d) and (e) to capture the small variation shown in these signals.

Holocene, suggesting these signals are still affected to a high degree by impurity localisation. Ice with larger grain sizes return
signals (collected under the same experimental conditions) with larger MADs with the RECAP Holocene ice showing larger MAD values for all signals.

The asymptotic behavior of the MAD plots motivates calculating the number of profiles required to improve the MAD by some factor. Table 2 contains the additional number of profiles required to achieve a relative decrease in MAD by a factor of two. These values show that large *relative* improvement in MADs can be made by measuring a small number of extra profiles.
Where an *absolute* MAD is targeted, thresholds such as the red line in each panel of Fig. 8, which illustrates reaching an arbitrary limit of 20 %, can be considered. The number of profiles required to reach this threshold is also recorded in Table 2.

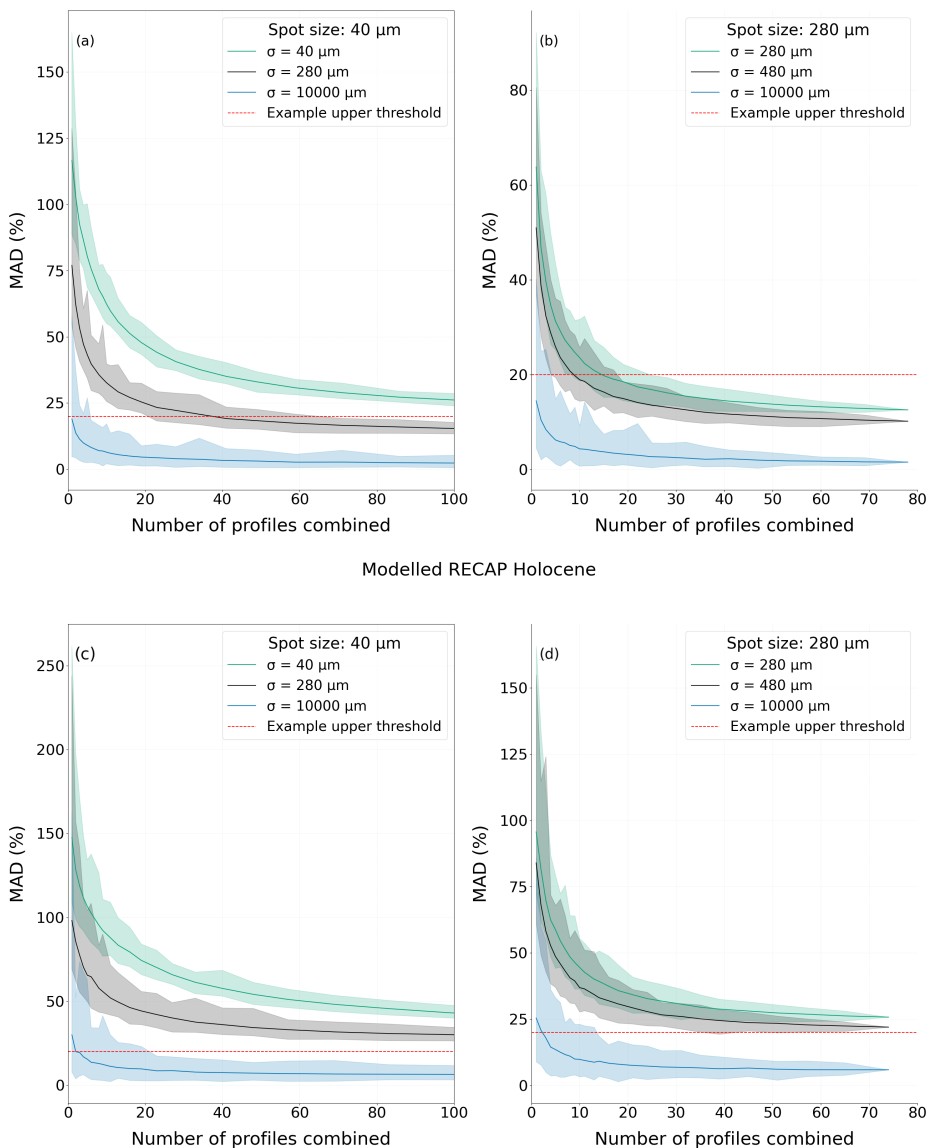

**Figure 8.** Plots of calculated MAD values against the number of LA-ICP-MS profiles used to construct a spatially averaged signal for the modelled EDC Holocene and RECAP Holocene faces shown in Fig. 6. As there are multiple ways to choose profiles for combination into a spatially averaged signal, the solid line of each colour shows the mean result and the shaded region shows the range of MADs acquired for different possible combined profiles. Panels (a) and (c) show results from simulating a 40 µm laser spot and (b) and (d) a 280 µm laser spot. Different coloured regions show MAD values resulting from smoothing with different width Gaussian kernels. An arbitrary threshold of 20 % is also shown (red line).

**Table 2.** Information on modelled sample signal representativeness. MAD values are a tabulation of key results calculated using equation 3 and are reported as the average MAD for the conditions indicated. MAD values for spatially averaged signals representing experimental conditions are calculated based on the number of profiles reported in Table 1. The best-case LA-ICP-MS MAD values apply to the specific sample geometry reported and assumes the entire width of the $2\,cm$ face is measured.

| Ice core | EDC | EDC | RECAP | RECAP |
|---|---|---|---|---|
| Climate period | Holocene | LGP | Holocene | LGP |
| MAD values (%) for experimental LA-ICP-MS conditions reported in Table 1 | | | | |
| Single profiles, $40\,\mu m$, unsmoothed | 117 | 135 | 146 | 130 |
| All profiles spatially averaged, $40\,\mu m$, unsmoothed | 62 | 78 | 112 | 85 |
| All profiles spatially averaged, $40\,\mu m$, $\sigma = 10\,000\,\mu m$ | 6 | 9 | 15 | 11 |
| CFA MAD value (%) | | | | |
| Smoothed with $\sigma = 10\,000\,\mu m$ kernel | 0.7 | 0.7 | 1.6 | 1.5 |
| MAD values (%) for all face profiles spatially averaged LA-ICP-MS | | | | |
| $40\,\mu m$, unsmoothed | 19 | 26 | 33 | 24 |
| $40\,\mu m$, $\sigma = 10\,000\,\mu m$ | 1.6 | 4.7 | 5.6 | 1.9 |
| $280\,\mu m$, unsmoothed | 15 | 19 | 26 | 18 |
| $280\,\mu m$, $\sigma = 10\,000\,\mu m$ | 1.5 | 4.7 | 5.9 | 1.9 |
| Approximate increase in number of LA-ICP-MS profiles required to reduce MAD by a factor of 2 | | | | |
| $40\,\mu m$, unsmoothed | 11 | 13 | 20 | 13 |
| $40\,\mu m$, $\sigma = 10\,000\,\mu m$ | 4 | 5 | 6 | 5 |
| $280\,\mu m$, unsmoothed | 5 | 5 | 8 | 5 |
| $280\,\mu m$, $\sigma = 10\,000\,\mu m$ | 4 | 5 | 5 | 5 |
| Total number of spatially averaged LA-ICP-MS profiles required for less than 20 % MAD | | | | |
| $40\,\mu m$, unsmoothed | 456 | [1]- | - | - |
| $40\,\mu m$, $\sigma = 10\,000\,\mu m$ | 1 | 2 | 3 | 2 |
| $280\,\mu m$, unsmoothed | 15 | 31 | - | 35 |
| $280\,\mu m$, $\sigma = 10\,000\,\mu m$ | 1 | 1 | 2 | 2 |

[1]- indicates the value is unreachable

## 4    Discussion

### 4.1    Measured impurity distribution

The empirically measured single-line profiles in Fig. 5 (a) and their modelled equivalent in Fig. 7 (a) show large intensity spikes with clear differences between profiles separated by even short distances on the ice's surface. This is the same behaviour shown by data collected during previous LA-ICP-MS studies (e.g. Bohleber et al., 2021) and is interpreted as an effect of the localisation of impurities at grain boundaries, as illustrated in Fig. 3. In this context, sodium provides a suitable archetypal soluble impurity with distribution mainly at grain boundaries, simplifying considerations of impurities located in grain interiors. Combining experimentally measured profiles to make spatially averaged signals resulting in output such as that plotted in Fig. 5 (b) and subsequent smoothing, shown in (c), results in signals with less variability.

Notably, as experimentation between ice samples was carried out on different days without calibration, intensity signals are not directly comparable between plots. Here, recently developed techniques for calibrating high-resolution LA-ICP-MS data (Bohleber et al., 2024) would allow for more straightforward comparisons to be made between data taken from different core sections over different time periods.

### 4.2    Model suitability to represent ice samples

With comparatively small maps (a few $mm^2$) as input and knowledge of the local average grain size, this new framework can generate a 3D ice volume representing the ice sample's structure and impurity distribution. By design, the model's representativity of physical ice samples holds in a statistical sense, justifying the transferability of model findings back to physical ice samples. Although a one-to-one physical representation was not the target, some noteworthy analogies exist: Voronoi tessellations are frequently used to create modelled structures representing polycrystalline materials such as metals (Zheng et al., 1996). There is a phenomenological link between grains in metallic systems and glacier ice, with both material classes growing over time according to a similar growth law (Alley et al., 1986), that motivates the use of Poisson Voronoi tessellations to form the microstructure of modelled ice samples. Poisson Voronoi tessellations, glacier ice, and metals have similar grain volume distributions and grain shapes. In ice, grain shapes are typically considered isometric (Cuffey and Patterson, 2010). It was found that a distance metric with p = 3 in equation 1 produces the most appropriate shape grains for modelling ice. Changing to Euclidean distance (p=2) produces solely planar grain boundaries, and increasing p above three increases computational time with little difference in grain shapes.

With grain shapes determined by the chosen growth model, the grain volume remains the only free parameter. The mean grain volume determines the appropriate number density of seed points in the Voronoi diagram required to produce target modelled grain volumes. Therefore, estimating a representative mean grain volume or radius for the ice sample plays an important role in generating an adequate representation in the model. For the space dimensions and grain radii reported in Table 1, the modelled spaces have grains only partially contained in the modelled volume with no full grains modelled, meaning the grain volumes can not be directly estimated. This clearly adds further complexity. A detailed discussion on the grain size distribution of

modelled volumes is contained in the supplementary material. This material shows that grain volumes vary around the mean grain volume conforming well to a gamma distribution. Notably, this grain volume distribution closely matches the empirically observed log-normal grain volume distribution from the EDC ice core for grains at depths above 2812 m (Durand et al., 2009). Normal grain growth dominates grain evolution over recrystallisation processes at these depths at the EDC drill site. Since there is no similar study available for the RECAP ice core, we assume this grain size distribution also suitably applies to these samples. Comparing the modelled EDC Holocene face, (a) and (b) in Fig. 6, with the RECAP Holocene face, (c) and (d), illustrates how the model captures different grain sizes and impurity imprints modelled for different samples.

## 4.3 Framework application

LA-ICP-MS ice core analysis is seeing growing interest, with several experimental setups being operated by different groups, all differing in experimental settings and spatial resolution. In this context, the framework presented here can allow improved comparison between the outputs of different experimental setups and can form a conceptual foundation for inter-technique comparisons, first and foremost with CFA, that can be further built upon.

The goal of many LA-ICP-MS analyses is to collect an underlying climate signal with 1D line profiles, the interpretation of which should be invariant of the method used to collect the signal and the lateral position from which data is collected. As recognised early on, it remains doubtful whether this goal has already been achieved (Della Lunga et al., 2017) and imaging fully revealed the origin of this problem lying in the grain boundary imprint (Bohleber et al., 2020). To contribute further to this discussion, the modelled ice volumes produced in this work can be analysed for a variety of different ice and impurity conditions and without the constraints placed on experimental analyses. While routine experimental LA-ICP-MS currently facilitates the collection of square millimetre sized maps and 10s of profiles from the surface of ice samples, modelled volumes can be used to construct square centimetre sized maps and all possible profiles throughout a 3D volume. While discussion confined to a 2D matrix would add value in itself, the 3D nature of the model allows the simulation of a CFA signal and therefore a direct comparison of modelled LA-ICP-MS and CFA signals, which should be verified against experimental data in the future, which is only possible as this is a 3D model. A consideration for experimental verification is that a direct comparison between signals generated by experimental LA-ICP-MS and CFA are currently not possible due to spatial offsets introduced between techniques during measurement. This offset arises as LA-ICP-MS measurements are carried out on the outer portion of ice that would be sent to waste as a decontamination procedure during CFA (Dallmayr et al., 2016).

The variability in experimentally acquired signals results from a superposition of different signals originating from the grain structure, ice layering, experimental settings and more. On the other hand, intentionally in its present configuration, any variation in modelled signals is due to impurity localisation, allowing this effect to be investigated in isolation. This variation is illustrated in Fig. 7, with all signals displaying some deviation from $\bar{I}$. The model allows quantification of the magnitude of this imprint across scales and makes quantitative predictions on the experimental design (e.g. how many line profiles to collect) required to manage this imprint. Notably, this application is independent of whether calibrated signals or intensities are analysed.

### 4.3.1 Capturing a representative signal

To date, criteria guiding the collection of layer-representative signals using LA-ICP-MS have been suggested based on the coherence of line profiles (Bohleber et al., 2021) and coherence with CFA signals (Spaulding et al., 2017; Della Lunga et al., 2017). In the modelled space employed here, a signal that fully captures the underlying layer would have the value $\bar{I}$ at all positions. Therefore signals which are coherent are likely also representative. From top to bottom, ((a) through (e)), the panels in Fig. 7 show convergence to $\bar{I}$. This convergence can be explained by each subsequent signal resulting from profiles that sample more volume per unit depth or have increasing smoothing between depths. Clearly, signals that sample more material per unit depth, such as CFA, are less influenced by variations due to the spatial distribution of impurities. Accordingly, it is not surprising that Table 2 shows that simulated CFA signals have the lowest MADs. Grain size is not the only factor affecting signal MADs, with the specific impurity distribution, influenced by the impurity species and climate period, and the localisation process itself forming important considerations.

Quantification of signal representativeness can be used to guide experimental design. Reported CFA MAD values are specific to a hypothetical melthead with a 1 by 2 $cm$ cross-section and would further reduce if a larger cross-sectional area (and therefore more volume per unit depth) is melted. Comparing MADs for different samples reveals a possible motivation for requiring CFA analysis with higher representativeness. It is evident that a significant contribution to higher signal MADs is increasing grain size. This suggests that quantifying signal representativeness may become relevant for discrete and continuous bulk analysis on samples with very large grain volumes, e.g. in deep ice, as even centimetre-sized samples may only contain small numbers of grains and their boundaries.

### 4.3.2 LA-ICP-MS experimental design

In the case of LA-ICP-MS, experimental design is also driven by resolution and representativeness requirements. The MAD of LA-ICP-MS signals is significantly reduced when smoothed to the same resolution as CFA, therefore increasing representativeness. This vertical resolution reduction can be achieved through a combination of measuring with larger spot sizes, using an analytical system with large vertical signal mixing, or applying smoothing in post-processing. Collecting the most representative LA-ICP-MS signals requires combining and smoothing all signals on the analysed face to give a MAD of 1.5 % for the EDC Holocene sample, approaching that of CFA. In cases where high vertical resolution is critical, signals with even lower MAD values can be collected by analysing a larger surface area, similar to increasing the cross-sectional area of ice analysed using CFA. This could be achieved by measuring samples with larger surface areas or by collecting profiles from a fresh, i.e. deeper in the X or Y plane, ice surface. However, this comes at the expense of increased measurement time.

The requirement that many profiles must be averaged to produce high-resolution LA-ICP-MS signals with high representativeness corroborates the assertion made by Della Lunga et al. (2017), that "the averaging of the LA[-ICP-MS] signal between two or more parallel tracks spaced by a few millimetres is not only desirable, but necessary". The asymptotic behaviour of the MADs, shown in Fig. 8, shows that increasing the number of profiles combined into a spatially averaged signal initially returns

a large reduction in MADs, and therefore increase in signal representativeness. Table 2 shows that a *relative* increase in the representativeness by a factor of 2 can be achieved by measuring an extra 11 profiles at $40\,\mu m$ spot size for the EDC Holocene samples, and that even fewer are required to achieve the same gain with larger spot sizes and more signal smoothing.

While relative improvements are useful benchmarks, experimental design should also consider the absolute target representativeness required to capture a climate signal, the criteria for which will depend on the depth, age, and estimated layer thickness of the target ice. To showcase a concrete example of how the model can be used to set an experimental design according to a predefined limit for tolerable signal MADs, we consider an arbitrarily selected MAD of 20 % acceptable. However, this is not a set value and should be adapted according to the specific objectives of a set of measurements. For EDC Holocene Table 2 shows this can be achieved through collecting signals in the following ways:

- at a resolution of $40\,\mu m$ with no signal smoothing, at least 465 profiles must be collected

- at a resolution of $280\,\mu m$ with no signal smoothing, at least 9 profiles must be collected

- at a resolution of $280\,\mu m$ with signal smoothing to CFA resolution, at least 1 profile must be collected

As the spatial distribution of impurities varies between elemental species and climate period, these values will vary for different ice core samples and impurity types. Notably the collection of an unsmoothed spatially averaged profile with a MAD of less than 20 % is not possible for the modelled RECAP Holocene sample, but illustrates the need for either many profiles or low vertical resolution to achieve representativeness. Therefore through determining a target MAD and depth resolution the nature of the analysis, e.g. LA-ICP-MS or CFA, can be set to best extract a layer-representative signal at the required resolution.

Under the right conditions LA-ICP-MS analysis can return signals with higher resolutions and similar representativeness to those produced using CFA. This positions LA-ICP-MS well as a tool to extract high-resolution climate signals, with the important added value of LA-ICP-MS being a micro-destructive technique, allowing revisiting of ice archives and performance of round-robin experiments among different laboratories with their own analytical strengths. Given the requirement for many profiles to be measured, there is a clear benefit in increasing the spatial extent over which information is collected by LA-ICP-MS. To achieve such measurements experimental and analytical developments are required. Large sample chambers have merits (Sneed et al., 2015; Stoll et al., 2023), and this should be considered during the cutting and processing of target ice samples. However, imaging areas larger than a few square millimetres currently requires prohibitively long measurements. This restriction can be somewhat mitigated by high repetition rate LA systems which may allow chemical data to be collected over very large surface areas of ice samples. The model developed here may significantly aid the design of such experiments, by a priori determining the desired spot size and resolution. This will allow representative, lateral position-invariant, signals to be collected at high-resolution using LA-ICP-MS. Yet, especially for deep ice, we will require a better understanding of how a climate signal manifests at the microscale. This involves a better understanding of the processes driving the localisation of soluble impurities at grain boundaries, possibly occurring during the transition from snow to firn and subsequently ice (Stoll et al., 2023). Related effects comprise impurity diffusion (Barnes et al., 2003; Ng, 2021), the study of which could utilise modelled ice structures created by 3D Poisson Voronoi tessellations.

### 4.3.3 Calibrated signals

Using calibrated LA-ICP-MS data as an input to this framework can strengthen its use cases and the links between LA-ICP-MS and CFA. Given the challenges involved in collecting calibrated LA-ICP-MS data (Miliszkiewicz et al., 2015; Mervič et al., 2024) a dataset comprising both calibrated 2D and cm-length 1D LA-ICP-MS data on ice core samples is not currently available. However, a recent publication by Bohleber et al. (2024) details a calibration scheme and presents calibrated 2D data collected from ice samples originating close in depth to the EDC data discussed in this study.

To extend this discussion to incorporate this limited calibrated data, models were generated based on data collected from the EDC core at depths of 281.8 m (Holocene) and 1,096.5 m (LGP), as reported and published by Bohleber et al. (2024). Given the proximity of these samples to those reported in Table 1, the grain radii reported in this table are considered suitably representative. The experimental and modelled data the following discussion is based on, with data equivalent to Figures 4, 6, 7, and 8, can be found in Sect. S3 of the supplementary material, with the notable omission of experimental LA-ICP-MS profiles which were not measured.

The main effect of the calibration is to reduce the distance between the grain interior and boundary distributions, shown uncalibrated in Fig. 4, along the x-axis. This reduces the magnitude of variability between grain boundary and interior distributions. The impact of the change from uncalibrated to calibrated data on signal representativeness can be investigated in isolation of grain size variability by comparing the pairs of uncalibrated and calibrated Holocene and LGP data. Reducing variability between the grain boundary and interior regions reduces the variability in measured signals and, therefore, reduces signal MADs for all cases. Whether models and therefore conclusions on representativeness should be explored based on calibrated or uncalibrated data should be carefully considered. The range of intensities output by an ICP-MS instrument is large to allow high sensitivity and the shift to calibrated data is simply a transformation of this measured intensity. Calibrated signals should be measured experimentally, using both LA-ICP-MS and CFA, and the model's predictions validated empirically.

This calibrated data, and the 3D nature of the model, also allow comparison of the framework against further existing empirical data. The study by Bohleber et al. (2024) discusses LA-ICP-MS data in relation to calibrated bulk measurements. While the measurements return average concentrations similar in their order of magnitude, there are discrepancies between 2D maps and bulk data. Concentrations reported by Bohleber et al. (2024) for 2D LA-ICP-MS analys and bulk measurements are reported alongside new modelling results in Tab. 3.

The modelled bulk concentrations again agree to an order of magnitude with measured results. The variability in these values could be due to several factors, including 2D maps not fully representing bulk impurity content, spatial offsets in measurements, and grain size variability within a sample. This small-scale variability can now be investigated through a modelled sensitivity analysis, where modelled volumes with variability in grain sizes and impurity distribution have their bulk concentrations compared. This allows quantitive exploration of the small variability between modelled and measured bulk concentrations, and LA-ICP-MS map averages. These observations again motivate the collection of further calibrated datasets, including LA-ICP-MS and associated CFA data to extend this discussion from a bulk to a depth-wise comparison.

**Table 3.** Compilation of measured and modelled concentrations. The modelled prediction is the mean intensity of all voxels in the space. Note that the Holocene LA-ICP-MS and bulk measurements come from samples horizontally separated by 50 cm, as reported by Bohleber et al. (2024).

|  | Climate period | Holocene | LGP |
|---|---|---|---|
|  | 2D LA-ICP-MS map | 15 | 54 |
| Concentrations (ppb) | Bulk measurement | 30 | 35 |
|  | Modelled prediction | 15 | 30 |

## 4.4 Potential extensions

The framework presented can be adapted to a broad range of ice samples. In particular for deep ice, and at sites where ice is deformed under high temperatures and stresses such as at the site of the EGRIP core (Stoll et al., 2024), implementing additional constraints will be crucial. The structure and impurity distribution of ice must be suitably captured at the grainscale, which is much larger in deep ice. This entails simulation of signals over a large enough volume, which requires careful management of computational resources. The current process limiting computational performance is the speed of structure generation, which rapidly increases with increasing grain size in the current implementation. This explains the large increase in model generation time for the RECAP Holocene sample in comparison with the other samples. To overcome this limitation a more efficient structure generation could be implemented which exploits parallel processing. Then, the following applications appear as worth-while additions to improve the model representation of various ice conditions.

The modelled ice generated in this study represents the basic structure of ice well but does not include features typical of glacier ice beyond grains and their boundaries. Ice samples contain other prominent features in their microstructure such as bubbles and insoluble impurities. Work that characterises such features can be used to amend this framework to include their effects on impurity distribution (Bohleber et al., 2023; Stoll et al., 2021; Bendel et al., 2013). Considering insoluble inclusions will allow this framework to be extended to chemical species mostly present in dust. Profiles collected measuring the insoluble impurity component are not treated in the present study and likely show different representativeness behaviours. The probability distributions describing the localisation of these elements will likely be unique for each element and therefore dimensionality reduction techniques could be useful to allow analysis to be carried out to bring generalised insight into multiple impurities in one model.

To capture ice microstructure representative of that seen in deeper ice, grain shapes representing ice that has undergone re-crystallisation effects beyond normal grain growth will have to be generated. Ice subject to deformation undergoes dynamic recrystallisation processes that change the grain fabric (Cuffey and Patterson, 2010). The resulting structures are different to those generated by a Poisson Voronoi tessellation. A potential approach to creating such grain fabrics is to start with a microstructure such as a 3D Poisson Voronoi tessellation and model vertical ice deformation in uniaxial compression such as that seen as at ice domes. Modelling the microstructure evolution would yield a simplified combination of normal grain growth

and recrystallisation processes. There is extensive literature discussing the computational modelling of microstructure recrys-
tallisation (Hallberg, 2011). The redistribution of impurities under these recrystallisation processes and their impacts on the
recrystallisation processes themselves can also be incorporated into such a model. Ice with more complex microstructures can
also be modelled, for example, by implementing more sophisticated Voronoi tessellations to precisely capture grain volumes
(Simone et al., 2017), capture grain size transitions, (Bourne et al., 2020), and implement preferred growth directions (van
Nuland et al., 2021).

## 5   Conclusions

Attempts to extract paleoclimate signals with single line profiles measured by LA-ICP-MS have suffered from severe ambigu-
ities in the past. Combining many individual signals to produce a spatially averaged signal has been suspected as a potential
remedy, but only the framework developed here adds a quantitative dimension to this problem. To do so we employ a physical-
based model of the soluble microscopic ice chemistry constructed using empirical data collected with LA-ICP-MS and a 3D
model of the ice matrix represented by a Poisson Voronoi structure. The framework is designed to quantitatively assess the
imprint of the ice matrix, the grain boundary network, in 1D signals collected with LA-ICP-MS for various ice conditions.
These conditions are captured in samples analysed from both Greenland and Antarctica from both the Holocene and Last
Glacial Period. Results show that a spatially averaged signal resulting from the combination of all profiles on a modelled ice
sample's face varies on average by between 19 to 33 % in the presented cases, with increasing deviation for samples with
larger average grain sizes. This variation can be further reduced to between 1.5 and 5.9 % by smoothing these signals with a
Gaussian kernel to CFA resolution. The 3D nature of this model allows comparison between the surface LA-ICP-MS technique
and bulk meltwater analyses. Further additions to this framework are foreseen to extend the representation also to include in-
soluble impurities and a broader range of ice conditions. This framework provides a tool for quantitative guidance in setting
experimental parameters in LA-ICP-MS, which are important for both ice-impurity interactions and the layering commonly
associated with climatic signals. Considering the evident merits of LA-ICP-MS delivering high-resolution impurity signals in
a micro-destructive fashion, developments to this approach may become particularly crucial for the systematic planning of the
collection of data from deep ice samples. This especially concerns ice collected from the ongoing efforts to retrieve the oldest
continuous ice record from Antarctica and its interpretation.

*Code and data availability.* Datasets and code will be made publicly available via open-access repositories such as those facilitated by GitHub, Pangaea (www.pangaea.de) or Zenodo (www.zenodo.org) after acceptance of the manuscript.

*Author contributions.* Experimental measurements were designed and conducted by PL, PB, and NS. Software was written by PL with conceptualisation and support provided by PB and RR. The scope of the manuscript was developed by PL, RR, and PB and an initial manuscript draft was produced by PL. All authors contributed to the discussion of the results and the final version of the manuscript.

*Competing interests.* The authors declare that they have no conflict of interest.

*Acknowledgements.* The authors thank Sebastiano Vascon, Luca Palmieri, and Marcello Pelillo for their help with computational problems and for access to their computing power, likewise thanks go to Alessandro Bonetto, Ciprian Stremtan, and Stijn van Malderen for their continued technical support. This publication was generated in the frame of the DEEPICE project. The project has received funding from the European Union's Horizon 2020 research and innovation programme under the Marie Sklodowska-Curie grant agreement no. 955750.
The project has received funding from the European Union's Horizon 2020 research and innovation programme under grant agreement No. 815384 (Oldest Ice Core). It is supported by national partners and funding agencies in Belgium, Denmark, France, Germany, Italy, Norway, Sweden, Switzerland, The Netherlands and the United Kingdom. Logistic support is mainly provided by ENEA and IPEV through the Concordia Station system. This is Beyond EPICA publication number 41. Pascal Bohleber gratefully acknowledges funding from the European Union's Horizon 2020 research and innovation program under the Marie Skłodowska-Curie grant agreement no. 101018266.
Nicolas Stoll and Pascal Bohleber gratefully acknowledge funding from the Programma di Ricerche in Artico (PRA). Co-funded by the European Union (ERC, AiCE, 101088125). Views and opinions expressed are however those of the authors only and do not necessarily reflect those of the European Union or the European Research Council. Neither the European Union nor the granting authority can be held responsible for them.

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
