# Peer review of "What does the impurity variability at the microscale represent in ice cores? Insights from a conceptual approach"

_EGUsphere, 2024_

## Author Comment (AC1)

Review of manuscript entitled "What does the impurity variability at the microscale represent in ice cores? Insights from a conceptual approach" submitted to EGUsphere (TC) by Piers Larkman.

General comments:

This manuscript investigates the impact of measurement scale by comparing high-resolution 2D data from LA-ICP-MS with smoothed and converted 1D profiles aligned with the CFA measurement scale. High-resolution measurements are crucial for extracting climate signals from extremely thin layers of deep ice. Results from LA-ICP-MS analysis revealed that impurities, particularly sodium, are preferentially concentrated at grain boundaries rather than within the grain interiors. This indicates that impurity distribution becomes more heterogeneous as grain size increases. Consequently, signal deviation in 1D profiles (quantified as mean absolute deviation, MAD) increases with larger grain samples.

For most glaciologists, the CFA technique offers sufficiently high resolution; however, the resolution of tens of microns provided by LA-ICP-MS is an impressive advancement. Identifying the precise location of impurities is essential for understanding microstructures and deformation mechanisms.

The manuscript is well-written, well-organized, and could make a valuable contribution to the chemical analysis of ice cores and the oldest ice core project. Therefore, I recommend that the manuscript be accepted after minor revisions. I have some questions, comments, and suggestions regarding the results and their interpretation, which may require further explanation. Additionally, some figures need to be redesigned and modified to ensure they are easily understood by the readers.

We thank the referee for their insightful review, the reviewer's comments are addressed below in blue, alongside the original review text.

Specific comments:

The present manuscript focuses on sodium, which tends to accumulate at grain boundaries. In contrast, certain ions, such as chloride, dissolve and substitute for $H_2O$ molecules within the grain interior. In these cases, I suspect that the impact of grain size on measurement resolution (or the difference between 1D profiles and CFA) may not be as pronounced compared to impurities that accumulate at grain boundaries.

We agree that considering the specific microstructural location of different species in isolation is important. Based on existing data so far, we find that many elemental impurities are located at the grain boundaries, for which we take Na as an archetypical example case. Other, primarily insoluble and dust-related impurities, can also be found in the grain interiors, and these species are so far not considered in this approach. With ongoing developments to LA-ICP-MS allowing the measurement of more chemical channels, we plan to extend this work in more detail in the future.

We refer to the archetypical nature of Na in the introduction and this is stressed in section 4.1 of the revised manuscript. Furthermore, we have stressed that insoluble impurities in grain interiors are not directly targeted in this approach so far in sections 1 and 4.1 of the revised manuscript.

In Figure 4, impurities appear to be distributed within the grain interiors in both EDC and RECAP LGP samples. (In Figure 3, sodium distribution does not seem to extend into the grain interiors.) While this may not be the primary focus of the current paper, the differences in impurity distribution between glacial and interglacial ice samples are intriguing.

The authors concluded that signal deviation increases with larger grain sizes. I wonder whether this deviation is influenced solely by grain size. It seems likely that impurity type (whether substituted within the grain interior or concentrated at grain boundaries) and climate period (glacial or interglacial ice) could also play a role in signal deviation. If the authors have insights on these factors, I suggest them to share their perspectives, as it would be beneficial for readers.

We agree that multiple factors must be considered when attempting to explain the observed impurity localisation at grain boundaries and that these factors may vary in their relative magnitude among different climatic periods. A growing body of work (e.g. Della Lunga et al., 2017, Stoll et al., 2023, Bohleber et al., 2022) illuminates the impurity localisation in the microstructure for different climate periods featuring different grain sizes (Holocene, Younger Dryas, Glacial Stadials and Interstadials). An explanation and improved understanding of the origin of the impurity localisation remains an ultimate target. Here, we are focusing on exploring the implication of localisation for interpreting LA-ICP-MS line profiles and CFA profiles. Figure 4 primarily shows that grain boundaries consistently feature higher intensities with respect to the grain interiors – again using Na as an example case for a specific (but large) subset of impurities. As the implemented framework considers both changes in grain size and the specific impurity localisation, modelled signal deviation captures at least these two factors.

Additional factors influencing signal deviation have been expanded on in the discussion in section 4.3.1 of the revised manuscript. The topic of different impurity types has also been included in the revised manuscript's introduction.

The discussion in Section 4.4 *Potential extensions* is particularly significant. As the authors mentioned, one of the key future goals of this study is to accurately extract climate signals from deep, thin ice. The deepest ice layers are subject to dynamic recrystallization due to the high-temperature environment, leading to a grain volume distribution that deviates from a gamma distribution. Moreover, migration recrystallization introduces small grains and creates complex grain boundaries, resulting in non-isometric grain shapes. Such complexities in grain boundaries and size distribution are observed not only in deep ice but also in ice samples deformed under high temperature and stress, such as those from the EastGRIP ice core. Replicating these intricate microstructures within an ice matrix model presents significant challenges.

We again agree with both the value and challenge of modelling complex microstructures, including in deep ice and sites such as EastGRIP. This manuscript introduces the modelling approach, providing a basis for further in-depth studies. We take this comment as further encouragement to explore how this model can be extended in this and other regards in the future.

A note has been included in section 4.4 on cores (specifically EGRIP) originating from sites where high deformation is present.

Although it may be considered future work, the manuscript's discussion and practical implications could be greatly enhanced by including results and assessments from ice samples

with complex microstructures, such as those containing small grains and complicated grain boundaries formed by migration recrystallization. In my view, even without 3D ice matrix modeling, simply comparing LA-ICP-MS data with smoothed and combined 1D profiles (like Figure 5), and grain size (distribution) offers substantial value.

In principle, this is an excellent idea, and we hope to extend this work accordingly in the future. We must emphasize the substantial experimental effort in capturing the 2D impurity maps, although the technique is still rapidly evolving to measure more data faster. At present, the limited availability of LA-ICP-MS data prevents the inclusion of such a discussion in this manuscript. We take the suggestion as encouragement for future work, as noted.

L7 in the abstract: What does "high-frequency component of signals" mean?

This phrase was used to refer to the variations in 1D signals that change rapidly (in the case of the 1D signals discussed in the paper, the 'high-frequency' component arises due to the grain-boundary imprint causing rapid changes in intensity. By contrast, the 'low-frequency' component would be the slower changes due to e.g. a climate signal).

Given that the paper does not discuss variations in terms of frequency elsewhere, this sentence has been rewritten.

L15 in the abstract: *This approach guides collecting layer-representative signals from LA-ICP-MS line profiles and may help to bridge the scale gap between LA-ICP-MS data and data collected from meltwater analysis.*

As mentioned, this approach could assess the scale gap. On one hand, how does this approach specifically bridge the scale gap? I think this is the most interesting for general glaciologist.

The scale gap concerns the different spatial resolutions of the data generated by CFA and LA-ICP-MS, centimetres and micrometres, respectively. In the presence of a regularly cm-spaced stratigraphy encoding climate variability, there is a question of how this signal manifests in micron-resolution LA-ICP-MS data, which is highly influenced by the impurity-localisation at grain boundaries and thus a second factor, grain size. We try to elucidate this question in the present work, specifically for different grain size conditions and climatic periods. Before, the scale gap was mainly addressed heuristically by smoothing LA-ICP-MS data until they resemble existing CFA profiles (e.g. Della Lunga et al., 2017,  Bohleber et al 2021). Assistance from the model allows the scale gap to be bridged independently of comparison to CFA. It allows us to determine how different LA-ICP-MS profiles can capture cm-scale variability. The model allows studies that are not (easily) possible with empirical measurements, such as the generation of spatially coherent measurements with different experimental techniques and parameters, the isolation of components contributing to measured signals (e.g. due to impurity localisation or grain size).

The above discussion has been captured in the introduction of the revised manuscript.

L79 (Table 1) In the EDC ice samples, mean grain size in LGP is larger than that in Holocene? In Figure 3, grains in EDC LGP ice sample look small.

The area illustrated in figure 3 for LGP does indeed contain small grains; this is just a small snapshot of the surface measured using LA-ICP-MS. The optical/microstructure measurements of the grain sizes reported in table 1 are taken from and are collected over larger areas. This has been clarified in figure 3's caption. This example shows how important it is to obtain highresolution microstructural and laser ablation data to exploit the potential of the presented approach fully.

L63: What does "The computational representation uses the arguably most simple manifestation of a climate signal, a constant signal" mean? How does the climate signal mode affect the results and discussion of the present study such as experimental measurements and ice structure generation? Please provide brief explanation for general cryosphere's readers.

We realize that this needs further clarification. The main idea is that based on the cm-resolution of CFA, a CFA profile can be regarded as a sequence of discrete constant values spaced at the resolution of the system, 1 cm or up to 0.5 cm (the exact value may vary according to climatic period and local variation). This computational representation simplifies this signal to a discrete constant value, and we investigate under which conditions we can reliably extract this discrete constant value. Measuring an empirical signal under the conditions that extract a constant value from the model would arguably detect the cm-scale climate variability present in an ice sample if the (parallel) LA-ICP-MS profiles are extended along the main core axis.

We have clarified this point accordingly in section 2.1 where it was first mentioned.

L103: I understand that the modelling of 3D ice structure is useful. However, I didn't see how 3D ice structure helped the verification of results and discussion. Figures 4 to 8 indicate 1D or 2D results. 2D ice matrix model is not sufficient?

The 3D model is critical for the simulation of CFA signals. Based on additional feedback from reviewer 2, a discussion has been added regarding comparisons to bulk analysis, which also requires modelling a 3D structure. Points regarding the applicability of the 3D model have been stressed in the discussion.

L160: *A comparison of the optical images and intensity maps in Fig. 3 shows sodium is concentrated preferentially at the grain boundaries compared with grain interiors for all measured samples.*

Even shallow EDC sample (Holocene ice), sodium is concentrated grain boundaries. Does this mean that impurities are already concentrated at the grain boundaries during deposition?

This is a very interesting question, referring again to the physical cause(s) of the observed localisation. Due to the limited amount of data, we are not confident in attempting to answer this (yet). However, we can say that based on the data acquired thus far, the localisation of impurities such as Na appears to be already significant in upper ice sections. First results were shown by Stoll et al. (2023a) displaying that localisation in the NEEM ice core already occurs in the upper 50 m. Measuring firn with LA-ICP-MS remains challenging, but more research will be conducted to better assess the localisation process.

L188: *These modelled signals show the same general features as experimentally measured signals, with large spikes in intensity where profiles intersect grain boundaries.*

It is difficult to determine whether experimental and modelled profiles signals have similar behaviors. Replication of experimental results by means of a model is, in my view, important in the present study. Please provide a comparative figure between the experimental and modelled results.

The plots in figure 5 (a) and figure 6 (a) are equivalent for experimental and modelled results, this link has been highlighted in the results and stressed in the discussion.

Figure and table:

Table 1: Please provide explanation for "Profile lateral separation (mm)".

This value has been illustrated in figure 6 (in addition to the redesign suggested by both reviewers). The table 1 caption directs the reader to this illustration.

Figure 3: Redesign is required.

Three images (optical image grain boundary segmentation, and chemical map) are not identical at RECAP ice samples. For example, in Holocene sample, grains in the middle and right images are elongated vertically. In LGP sample, grains in the left image are elongated vertically. Please modify.

The optical image of EDC Holocene sample is low resolution, it is difficult to distinguish grain boundaries.

Size of EDC LGP samples is too small. Image sizes should be the same for all samples.

Figure 6: (a and c) It is difficult to see grain boundaries, please make contrast clearer.

(b and d) What do the color difference of grains mean?

Thank you for the suggestions, the figures have been redesigned accordingly. In fig. 6 b and d, the colour difference is arbitrary and holds no special meaning, the colour scheme has been updated and clarified in the figure caption.

Figure 7: Why do the smoothed profiles (d) and (e) appear to be different? (It doesn't even look similar to panels a to c)

This is also the case in Figures S6 and S9.

This phenomenon has been clarified in the revised manuscript in the figure's caption and in section 3.2. The deviations are very small from the modelled un-changing climate signal, on the order of 2% for LA-ICP-MS (7d) and 1% for CFA (7e), both modelled LA-ICP-MS and CFA signals collected from slightly different spatial locations can show varying trends in signals showing such small-amplitude variability. In this case, the modelling shows opposite trends greatly highlighting this effect. However, the modelled LA-ICP-MS and CFA signals are similar in that they only demonstrate minor deviation from the underlying signal. (This response is replicated in response to similar comments by both reviewers 1 and 2).

Figure 8: Bottom two panels are labeled with (a) and (b). Please modify. Additionally, the spot size in panel b and caption is shown as 280 um, but the main text explain 260 um. Please correct value.

The figure label and spot size disparity have been corrected (updated to 280 in the text).

References

Della Lunga, D., Müller, W., Rasmussen, S. O., Svensson, A., and Vallelonga, P.: Calibrated cryo-cell UV-LA-ICPMS elemental concentrations from the NGRIP ice core reveal abrupt, sub-annual variability in dust across the GI-21.2 interstadial period, The Cryosphere, 11, 1297–1309, https://doi.org/10.5194/tc-11-1297-2017, 2017

Stoll, N., Westhoff, J., Bohleber, P., Svensson, A., Dahl-Jensen, D., Barbante, C., and Weikusat, I.: Chemical and visual characterisation of EGRIP glacial ice and cloudy bands within, The Cryosphere, 17, 2021–2043, https://doi.org/10.5194/tc-17-2021-2023, 2023a.

Stoll N., Bohleber P., Dallmayr R., Wilhelms F., Barbante C., Weikusat I.: The new frontier of microstructural impurity research in polar ice. Annals of Glaciology 1–4. https://doi.org/10.1017/aog.2023.61, 2023b.

Bohleber, P., Stoll, N., Rittner, M., Roman, M., Weikusat, I., & Barbante, C.: Geochemical characterization of insoluble particle clusters in ice cores using two-dimensional impurity imaging. Geochemistry, Geophysics, Geosystems, 24, e2022GC010595. https://doi.org/10.1029/2022GC010595, 2022.

Bohleber, P., Roman, M., Šala, M., Delmonte, B., Stenni, B., and Barbante, C.: Two-dimensional impurity imaging in deep Antarctic ice cores: snapshots of three climatic periods and implications for high-resolution signal interpretation, The Cryosphere, 15, 3523–3538, https://doi.org/10.5194/tc-15-3523-2021, 2021.

---

## Author Comment (AC2)

This manuscript investigates the variability of impurity signals at the microscale by comparing high-resolution 2D impurity signals to modeled 1D CFA measurements. The authors seek to improve understanding of a significant issue within LA-ICP-MS analyses, namely that localized impurities, cause significant variation in 1D signals due to their uneven distribution across the ice matrix.

The model addresses a key challenge in interpreting impurity signals and offers a method to quantify the impact of impurity localization on 2D LA-ICP-MS signals. The manuscript is well written and offers guidance on the number of profiles and level of smoothing required to generate representative signatures. This information is critical for designing LA-ICP-MS experiments. However, while the manuscript provides a step forward in the experimental design, I have concerns regarding the level to which the modelled data accurately represents the climate signal and its applications.

We thank the referee for their encouraging comments and critical assessment of this manuscript. We appreciate the note that the model is useful for experimental design for LA-ICP-MS. The revised manuscript addresses the concerns regarding climate signal interpretation. The reviewer's comments are addressed below in blue, alongside the original review text.

The authors describe in the introduction that LA-ICP-MS analyses are necessary to reconstruct climate records in deep ice and use this as a primary reason for this study. However, the authors use the most basic structure of ice in this model. This is understandable given the continued questions around methodologies for LA-ICP-MS, however, as the work currently stands there are limited implications for deep ice studies.

This manuscript aims to step towards using LA-ICP-MS for high-resolution climate signal acquisition, such as that likely required for deep ice samples. It aims to demonstrate the presented model's applicability in a better-understood shallow ice regime before future work to model deep ice. We agree that the basic ice structure utilised does not reflect deep ice conditions and outline considerations to extend the discussion to deep ice.

The limitations of the current approach with regard to deep ice applications and potential additions for future work have been clarified in the introduction of the amended manuscript.

Additionally, while there is an attempt to quantify the CFA results using modeled results, no actual experimental data is provided to ground truth the model's ability to reconstruct CFA results. Additionally, as no concentrations are provided and no calibration was conducted, it is difficult to see whether the modeled data that this project hinges on are realistic or comparable. As a result I recommend the manuscript be reconsidered after major revisions.

Matrix-matched calibration is a widespread challenge in LA-ICP-MS, and only recently has significant progress been made in this regard for high-resolution studies on ice (Bohleber et al., 2024). In the future, we will target collecting and analysing additional calibrated LA-ICP-MS data. The presented modelling framework can be easily adapted to accept calibrated data. A comparison with experimental CFA data will also be targeted, as this will further test the model, as noted by the reviewer. Given that many studies using LA-ICP-MS to measure ice core samples use uncalibrated data (e.g. Della Lunga et al., 2014, Spaulding et al., 2017, Bohleber et al., 2021, Bohleber et al., 2022, Stoll et al., 2023), the present modelling discussion aims to contribute to the interpretation of analysis conducted on such signals. The routine use of calibrated data represents an exciting future development to extend both experimental and modelling studies.

Given the evident additional value calibrated data contributes to this topic, a discussion of recently-published calibrated ice LA-ICP-MS data that has a calibrated bulk data counterpart has been included to provide a ground-truth reference in the revised manuscript, as noted concerning the reviewer's comment on L222 below.

Specific comments:

L188: The statement "These modelled signals show the same general features as experimentally measured signals, with large spikes in intensity where profiles intersect grain boundaries." is not proven as the experimental signals are not shown for comparison.

This statement has been clarified to note that it refers to only the LA-ICP-MS profile data, plotted in figure 5, and not to modelled CFA, which does not have an experimental counterpart.

Figure 6: greater contrast is needed to see grain boundaries in a) and c). It is also very difficult to see the blue and red lines. Please make these thicker or choose different colors.

The colours used in the figure have been revised to make the features clearer.

Figure 7: There is no explanation I can find for why 7d and 7e show opposite profiles. Please provide more information. This is particularly important for the author's claim that this model can be used to compare between LA-ICP-MS and CFA results.

This phenomenon has been clarified in the revised manuscript in the figure's caption and in section 3.2. The deviations are very small from the modelled un-changing climate signal, on the order of 2% for LA-ICP-MS (7d) and 1% for CFA (7e), both modelled LA-ICP-MS and CFA signals collected from slightly different spatial locations can show varying trends in signals showing such small-amplitude variability. In this case, the modelling shows opposite trends greatly highlighting this effect. However, the modelled LA-ICP-MS and CFA signals are similar in that they only demonstrate minor deviation from the underlying signal. (This response is replicated in response to similar comments by both reviewers 1 and 2).

Line 222: I'm unclear why no calibration is used here. Particularly for comparing LA-ICP-MS to CFA signals or comparing between analyses, and ground truthing the model, this is important to understand how well the parameterized model works.

As noted above, we agree that calibration provides valuable insight. Based on the limited calibrated data available in the manuscript by Bohleber et al., 2024, a ground truth example in comparison to bulk data has been included in the revised manuscript's discussion section 4.3.3. This discussion presents ice modelled based on calibrated LA-ICP-MS data and compares the results to calibrated bulk analysis. These results and the reviewer's comments encourage future analysis to be carried out in comparison with CFA.

Line 254: "In this context, the framework presented here can allow improved comparison between the outputs of different experimental setups and can form an essential foundation for inter-technique comparisons, first and foremost with CFA." This has not been proven. The authors themselves mention that day-to-day comparisons are not comparable, and as no calibration or concentration data is provided this remains conceptual.

This statement has been revised to highlight the link is conceptual.

Line 264: "Furthermore, the simulation of a CFA signal allows a direct comparison of LA-ICP-MS and CFA signals which is only possible as this is a 3D model" This has not been shown in this paper as no comparison to experimental CFA data is provided to show these are comparable.

This statement has been revised to highlight this work provides the framework for such a comparison and that further (ongoing) work is required with calibrated data collected using both techniques to satisfy a link.

Line 264: What does "This facilitates a direct comparison that is not currently possible for physical ice samples as the outer portion of ice measured using CFA is not measured to avoid contamination (Dallmayr et al., 2016)." Mean? I'm unclear how direct comparison is only possible with a 3D model here and why contamination control procedures impede this.

This statement was intended to illustrate the fact that typical surface LA-ICP-MS and bulk CFA analyses can not be carried out on the same part of a sample and that there will be spatial offsets in their measurement. As the outer portion of a CFA stick's cross-section is usually fed to waste, and LA-ICP-MS measures a surface, they do not measure the same part of a sample. This has been clarified in the revised manuscript.

References

Bohleber, P., Larkman, P., Stoll, N., Clases, D., Gonzalez de Vega, R., Šala, M., Roman, M., and Barbante, C.: Quantitative insights on impurities in ice cores at the micro-scale from calibrated LA-ICP-MS imaging, Geochem. Geophys. Geosyst., 25, https://doi.org/10.1029/2023GC011425, 2024.

Stoll N., Bohleber P., Dallmayr R., Wilhelms F., Barbante C., Weikusat I.: The new frontier of microstructural impurity research in polar ice. Annals of Glaciology 1–4. https://doi.org/10.1017/aog.2023.61, 2023.

Bohleber, P., Stoll, N., Rittner, M., Roman, M., Weikusat, I., & Barbante, C.: Geochemical characterization of insoluble particle clusters in ice cores using two-dimensional impurity imaging. Geochemistry, Geophysics, Geosystems, 24, e2022GC010595. https://doi.org/10.1029/2022GC010595, 2022.

Bohleber, P., Roman, M., Šala, M., Delmonte, B., Stenni, B., and Barbante, C.: Two-dimensional impurity imaging in deep Antarctic ice cores: snapshots of three climatic periods and implications for high-resolution signal interpretation, The Cryosphere, 15, 3523–3538, https://doi.org/10.5194/tc-15-3523-2021, 2021.

Spaulding, N., Sneed, S., Handley, M., Bohleber, P., Kurbatov, A., Pearce, N., Erhardt, T., and Mayewski, P.: A new multielement method for LA-ICP-MS data acquisition from glacier ice cores, Environ. Sci. Technol., 51, 13 282–13 287, https://doi.org/10.1021/acs.est.7b03950, 2017.

Della Lunga D., Müller W., Rasmussen S.O., Svensson A.: Location of cation impurities in NGRIP deep ice revealed by cryo-cell UV-laser-ablation ICPMS. *Journal of Glaciology*. 60(223):970-988, https://doi.org/10.3189/2014JoG13J199, 2014.

---

## Author Response (AR1)

**Response to editor decision**

Dear Nanna B. Karlsson

On behalf of myself and my co-authors, thank you for the invitation to upload a version of the manuscript revised based on reviewers' comments.

This revised version addresses the comments made by the reviewers. These changes are detailed below, directly following the responses shared in the interactive discussion, with specific detail on which additions have been made to address the reviewer's comments.

The revised manuscript has been uploaded, alongside a version with tracked changes. A revised supplement has also been provided.

I look forward to hearing back from you regarding the manuscript.

Kind regards

**Piers Larkman**

Relevant changes based on reviewer comments. Excerpts from the review comments are in black, revisions are shown in blue. **Line numbers refer to the version of the revised manuscript with visible tracked changes**.

**Amendments made based on to review 1**

The present manuscript focuses on sodium, which tends to accumulate at grain boundaries. In contrast, certain ions, such as chloride, dissolve and substitute for $H_2O$ molecules within the grain interior. In these cases, I suspect that the impact of grain size on measurement resolution (or the difference between 1D profiles and CFA) may not be as pronounced compared to impurities that accumulate at grain boundaries.

The archetypical nature of Na as a soluble impurity is further noted on L65, and L246-247.

The fact that further considerations are required for modelling insoluble impurities in grain interiors is noted on L71, and L247

The authors concluded that signal deviation increases with larger grain sizes. I wonder whether this deviation is influenced solely by grain size. It seems likely that impurity type (whether substituted within the grain interior or concentrated at grain boundaries) and climate period (glacial or interglacial ice) could also play a role in signal deviation. If the authors have insights on these factors, I suggest them to share their perspectives, as it would be beneficial for readers.

Additional factors influencing signal deviation have been listed on L315 - 317

The variability between different impurity types has been noted on L42

Such complexities in grain boundaries and size distribution are observed not only in deep ice but also in ice samples deformed under high temperature and stress, such as those from the EastGRIP ice core.

A note has been included on L409-410, specifically referencing EGRIP, noting high deformation sites require further consideration.

L7 in the abstract: What does "high-frequency component of signals" mean?

This term has been revised on L7 of the abstract

L15 in the abstract: *This approach guides collecting layer-representative signals from LA-ICP-MS line profiles and may help to bridge the scale gap between LA-ICP-MS data and data collected from meltwater analysis.*

As mentioned, this approach could assess the scale gap. On one hand, how does this approach specifically bridge the scale gap? I think this is the most interesting for general glaciologist.

In the introduction of the revised manuscript, the scale gap discussion has been introduced on L44, and stressed on L57-59

L79 (Table 1) In the EDC ice samples, mean grain size in LGP is larger than that in Holocene? In Figure 3, grains in EDC LGP ice sample look small.

This has been clarified in figure 3's caption.

L63: What does "The computational representation uses the arguably most simple manifestation of a climate signal, a constant signal" mean? How does the climate signal mode affect the results and discussion of the present study such as experimental measurements and ice structure generation? Please provide brief explanation for general cryosphere's readers.

Section 2.1 has been reworked to clarify this. Specifically the paragraph at L83 – 88 now introduces this idea

L103: I understand that the modelling of 3D ice structure is useful. However, I didn't see how 3D ice structure helped the verification of results and discussion. Figures 4 to 8 indicate 1D or 2D results. 2D ice matrix model is not sufficient?

The added discussion in the new section 4.3.3 relies on a 3D structure to predict bulk concentrations.

Points regarding the additional value of the 3D model have been added on L156, L294-295

L188: *These modelled signals show the same general features as experimentally measured signals, with large spikes in intensity where profiles intersect grain boundaries*.

It is difficult to determine whether experimental and modelled profiles signals have similar behaviors. Replication of experimental results by means of a model is, in my view, important in the present study. Please provide a comparative figure between the experimental and modelled results.

The plots in figure 5 (a) and figure 7 (a) are equivalent for experimental and modelled results. This is highlighted in the caption for figure 7, and on L244

Figure and table:

Table 1: Please provide explanation for "Profile lateral separation (mm)".

This value has been illustrated in figure 6 (in addition to the redesign suggested by both reviewers). The table 1 caption directs the reader to this illustration.

Figure 3: Redesign is required.

Three images (optical image grain boundary segmentation, and chemical map) are not identical at RECAP ice samples. For example, in Holocene sample, grains in the middle and right images are elongated vertically. In LGP sample, grains in the left image are elongated vertically. Please modify.

Aspect ratio and covered areas in figure 3 have been revised and the plot re-made

The optical image of EDC Holocene sample is low resolution, it is difficult to distinguish grain boundaries.

A smaller image has been provided to aid readability, although this image is still at low resolution

Size of EDC LGP samples is too small. Image sizes should be the same for all samples.

Plot sizes have been unified

Figure 6: (a and c) It is difficult to see grain boundaries, please make contrast clearer.

Contrast for this figure, and all similar, have been increased

(b and d) What do the color difference of grains mean?

The colour scheme has been updated and clarified in the figure caption.

Figure 7: Why do the smoothed profiles (d) and (e) appear to be different? (It doesn't even look similar to panels a to c)

This is also the case in Figures S6 and S9.

This is explained on L218 – 222, and the y-axis change stressed in the captions of figures 5 and 7

Figure 8: Bottom two panels are labeled with (a) and (b). Please modify. Additionally, the spot size in panel b and caption is shown as 280 um, but the main text explain 260 um. Please correct value.

This is updated on L 227

**Amendments made based on to review 2**

The authors describe in the introduction that LA-ICP-MS analyses are necessary to reconstruct climate records in deep ice and use this as a primary reason for this study. However, the authors use the most basic structure of ice in this model. This is understandable given the continued questions around methodologies for LA-ICP-MS, however, as the work currently stands there are limited implications for deep ice studies.

The requirement for extension for deep ice application is clarified on L 69 - 71

Additionally, while there is an attempt to quantify the CFA results using modeled results, no actual experimental data is provided to ground truth the model's ability to reconstruct CFA results. Additionally, as no concentrations are provided and no calibration was conducted, it is difficult to see whether the modeled data that this project hinges on are realistic or comparable. As a result I recommend the manuscript be reconsidered after major revisions.

Section 4.3.3. has been added that adds further ground-truth examples, including reference to calibrated data

Specific comments:

L188: The statement "These modelled signals show the same general features as experimentally measured signals, with large spikes in intensity where profiles intersect grain boundaries." is not proven as the experimental signals are not shown for comparison.

This statement has been clarified on L210- 211

Figure 6: greater contrast is needed to see grain boundaries in a) and c). It is also very difficult to see the blue and red lines. Please make these thicker or choose different colors.

The colours used in the figure have been revised to make the features clearer.

Figure 7: There is no explanation I can find for why 7d and 7e show opposite profiles. Please provide more information. This is particularly important for the author's claim that this model can be used to compare between LA-ICP-MS and CFA results.

This is explained on L218 – 222, and the y-axis change stressed in the captions of figures 5 and 7

Line 222: I'm unclear why no calibration is used here. Particularly for comparing LA-ICP-MS to CFA signals or comparing between analyses, and ground truthing the model, this is important to understand how well the parameterized model works.

The added section 4.3.3. discusses this work in the context of limited available calibrated data

Line 254: "In this context, the framework presented here can allow improved comparison between the outputs of different experimental setups and can form an essential foundation for inter-technique comparisons, first and foremost with CFA." This has not been proven. The authors themselves mention that day-to-day comparisons are not comparable, and as no calibration or concentration data is provided this remains conceptual.

This statement has been revised on L284

Line 264: "Furthermore, the simulation of a CFA signal allows a direct comparison of LA-ICP-MS and CFA signals which is only possible as this is a 3D model" This has not been shown in this paper as no comparison to experimental CFA data is provided to show these are comparable.

This statement has been revised on L294

Line 264: What does "This facilitates a direct comparison that is not currently possible for physical ice samples as the outer portion of ice measured using CFA is not measured to avoid contamination (Dallmayr et al., 2016)." Mean? I'm unclear how direct comparison is only possible with a 3D model here and why contamination control procedures impede this.

This has been re-addressed on L297 – 301

---

## Author Response (AR2)

**Response to editor decision received 06/01/2025**

Dear Nanna B. Karlsson

My co-authors and I thank you again for your communication. We have addressed the comments from referee #1, with our responses detailed below.

I have uploaded a new version of the manuscript, alongside a version with changes highlighted, and an updated version of the supplementary material.

We look forward to the next steps for the manuscript

Kind regards

Piers Larkman

**Response to Reviewer #1 –** reviewer text in black, response in blue line numbers refer to lines in the manuscript version with markup

Thank you for the revised manuscript and responses to the comments. The manuscript has been improved well. I agree with the addition of Section 4.3.3 (Calibrated signals). Calibration of experimental data is important for the practical application. However, I suggest some modifications (clarification) to the description in Section 4.3.3, Table 3 and S3 to make it easier for the reader to understand. Once these minor revisions are made, I would recommend it for acceptance.

Thank you for your comments, we agree that the revised manuscript was improved based on the previous round of review. We also agree that section 4.3.3, which discusses some calibrated data, was an important addition prompted by comments made by both first-round reviewers. In the context of improving this section, the reviewer's comments are addressed below in blue, alongside the review text.

Questions and suggestions for improvement

Please provide a short overview of the calibration in the present manuscript, even if it was detailed in Bohleber et al. (2024).

L371- 376 of the revised manuscript contains a short description of the calibration process implemented by Bohleber et al. (2024), which utilises small droplets with known elemental concentrations to produce a calibration curve which is subsequently applied to calibrate LA-ICP-MS maps.

L379:
What does "The main effect of the calibration is to reduce the distance between the grain interior and boundary distributions" mean? I can understand this reduces the variability magnitude between interior and boundary, and signal MADs. Why is it necessary to reduce the distance between the grain interior and boundary distributions?

The interpretation presented here is the intended interpretation. The revised manuscript removes the use of the word 'distance' and states on L384-385 that "The main effect of the calibration is to reduce the magnitude of variability between the grain interior and boundary distributions"

Table 3:
I think the explanation of Table 3 (caption and mention in the main text) is not enough. I did not understand the meaning of Table 3. Are the values in the table results from a specific samples or an overall average? Please explain how these values were estimated and what the values claim (differences in values depending on methods, and between LGP and Holocene).

The following points relating to the table have been clarified, with additions made to both the caption the written text. L396-397 states that the results are from specific samples and provide

only snapshots. The figure caption specifies the methods used by Bohleber et al. (2024) to arrive at the 2D LA-ICP-MS map concentrations, by averaging over pixels in the measured maps, and bulk measurements, through use of discrete ICP-time-of-flight-MS measurements. The original text on L400-404 notes that difference in values between methods could be due to a number of factors (including 2D maps not fully representing bulk impurity content, spatial offsets in measurements, and grain size variability within a sample) and illustrates the need for more research to unify impurity measurements across scales and dimensions. Discussing the relevance of measured difference in concentrations between the Holocene and LGP samples is beyond the scope of this study.

S3 Calibrated data results:
It would be useful to place the calibrated and uncalibrated data side by side to understand the effect of calibration. With the present manuscript, it is difficult to know how calibration will change the result.

We agree that comparison between the two sets of results – calibrated and uncalibrated – is important. Therefore figures S15 and S20 (a) have been updated to include the non-calibrated intensity scale alongside the calibrated data.